# Experimental warming and drying increase older carbon contributions to soil respiration in lowland tropical forests

Karis J. McFarlane [1] ✉, Daniela F. Cusack [2,3,4], Lee H. Dietterich [2,5,6], Alexandra L. Hedgpeth[1,3], Kari M. Finstad [1] & Andrew T. Nottingham [4,7]

Tropical forests account for over 50% of the global terrestrial carbon sink, but climate change threatens to alter the carbon balance of these ecosystems. We show that warming and drying of tropical forest soils may increase soil carbon vulnerability, by increasing degradation of older carbon. In situ whole-profile heating by 4 °C and 50% throughfall exclusion each increased the average radiocarbon age of soil $CO_2$ efflux by ~2–3 years, but the mechanisms underlying this shift differed. Warming accelerated decomposition of older carbon as increased $CO_2$ emissions depleted newer carbon. Drying suppressed decomposition of newer carbon inputs and decreased soil $CO_2$ emissions, thereby increasing contributions of older carbon to $CO_2$ efflux. These findings imply that both warming and drying, by accelerating the loss of older soil carbon or reducing the incorporation of fresh carbon inputs, will exacerbate soil carbon losses and negatively impact carbon storage in tropical forests under climate change.

Tropical forests exchange more $CO_2$ with the atmosphere than any other terrestrial biome[1], store nearly one-third of global soil carbon stocks[2], and have the highest soil $CO_2$ efflux of any ecosystem[3]. Tropical terrestrial ecosystems also have the shortest mean residence time for carbon on Earth, as short as 6–15 years[4,5], meaning that any change in carbon inputs or outputs could have large and relatively rapid consequences for tropical ecosystem carbon balance. Climate projections suggest a future that will be both warmer and drier for much of the tropics[6] with increasing drought intensity and dry season length for the Neotropics[7,8]. Despite the importance of tropical forests and their soils to the global carbon cycle and feedbacks to climate, uncertainty in predicting the response of tropical carbon cycling to future climate change remains high.

Soil $CO_2$ efflux is highly sensitive to temperature and moisture, which together have been shown to determine interannual patterns in emissions globally[9]. Even in tropical forests, where mean annual temperature is relatively high and temperature variability is relatively low, soil $CO_2$ efflux has been shown to increase with increasing temperature and peak at intermediate soil moisture content[10,11]. Meta-analyses of warming experiments across global terrestrial ecosystems have reported average increases in soil $CO_2$ efflux with warming of 9%[12]–12%[13], and reported increases in soil $CO_2$ efflux following whole-profile warming are even higher (e.g., 34%[14]–55%[15]). Thus, extrapolation of results from warming experiments suggests that climate warming will stimulate a net loss of global soil carbon to the atmosphere[16]. Importantly, none of the studies included in these meta-analyses were conducted in the tropics. Field warming experiments in tropical forests have only recently been instigated and early results show large increases in soil $CO_2$ efflux with increased temperature[15] as have laboratory incubations of tropical soils[17,18]. Soil moisture is also an important factor influencing soil microbial activity and respiration, and in the tropics, the seasonal variation in moisture is often greater

[1]Center for Accelerator Mass Spectrometry, Lawrence Livermore National Laboratory, Livermore, CA, USA. [2]Department of Ecosystem Science and Sustainability, Colorado State University, Fort Collins, CO, USA. [3]Department of Geography, University of California – Los Angeles, Los Angeles, CA, USA. [4]Smithsonian Tropical Research Institute, Panama City, Panama, Republic of Panama. [5]Department of Biology, Haverford College, Philadelphia, PA, USA. [6]Environmental Laboratory, U.S. Army Engineer Research and Development Center, Vicksburg, MS, USA. [7]School of Geography, University of Leeds, Leeds, UK. ✉e-mail: kjmcfarlane@llnl.gov

than that of temperature[10,19,20]. However, field drying experiments in tropical forests have reported mixed responses of soil $CO_2$ efflux to drying, including increases[21], decreases[22,23] and no responses[24] across forests of differing rainfall and seasonality. Field responses to drying have varied even among nearby forests, apparently related to baseline moisture and fertility[11].

Most of the previous work in tropical forests only considered total $CO_2$ efflux rates, which are important for determining the overall carbon balance of tropical forests[25], but are limited in their ability to uncover mechanisms behind observed change. Those mechanisms can be revealed by the determination of $^{14}C$ values, which indicate the average age of the carbon sources being metabolized and released as $CO_2$[26], where in this context 'new' or 'young' carbon has been fixed from the atmosphere in the last few years, older 'decadal-aged' carbon is enriched in $^{14}C$ relative to the current atmosphere, and even older 'century or millennial-aged' carbon is depleted in $^{14}C$ relative to current atmosphere (see section "Methods" and Supplementary Fig. 1). These studies have provided valuable information on the response of the soil carbon cycle to climate change across a range of in situ experiments. For example, in thawing permafrost, $^{14}C$ signatures of soil $CO_2$ efflux revealed that warming and drying together caused an increase in the release of old carbon (depleted in $^{14}C$ relative to atmosphere) in soil carbon as $CO_2$[27]. In a temperate conifer forest, whole-profile soil warming increased soil $CO_2$ efflux and decreased soil carbon stocks by about 30%, changes that were attributed to increased decomposition of decadal-aged soil carbon pools[14,28]. In a temperate deciduous forest where experimental drying decreased soil $CO_2$ efflux by 10–30%, $\Delta^{14}C$ of soil $CO_2$ efflux attributed this response to decreased microbial respiration near the soil surface[29]. In contrast, neither the rate nor the $\Delta^{14}C$ of soil $CO_2$ efflux were affected by experimental drying in a tropical forest in Tapajos, Brazil[24]. We have an extremely limited understanding of how tropical forest soil $CO_2$ efflux (or its $\Delta^{14}C$) are affected by warming and drying in the same forest. Given that temperature and moisture are the major climatic drivers of soil $CO_2$ efflux[9,10,15,19,20], and that in the tropics both significant warming and drying are predicted this century[30], there is a critical need for studies that assess the impact of warming and drying together on both the magnitude and source (i.e., age) of soil $CO_2$ efflux in tropical forests.

In this study, we determined how warming and drying impact the amount and age of carbon released as soil $CO_2$ efflux in two distinct lowland tropical forest areas. We measured the $\Delta^{14}C$ and $\delta^{13}C$ of soil-respired $CO_2$ in Panamanian forests (Fig. 1 and Supplementary Table 1) that are subject to either in situ experimental soil warming (4 °C above ambient temperature to 1.2 m depth[15,31]) or in situ experimental drying (50% throughfall exclusion[11,32,33]). Our study sites are seasonally moist, semi-deciduous forests. The warming site and one drying site are both within the Barro Colorado Nature Monument in nearby and similar forests on similar soils, enabling a direct comparison of warming and drying effects on soil $CO_2$ efflux. A second drying experiment is on the northern side of the Panama Isthmus on infertile soils where mean annual precipitation (MAP) is greater, representative of a broad geographic area of the tropics. Given the seasonality of these forests, we performed measurements at stages of the seasonal cycle for which we expected the largest variation in $CO_2$ efflux between control and experimental plots based on previous studies[11,15,20]—the wet season and dry season or dry-to-wet season transition (see section "Methods").

We show that warming and drying both increase the relative contribution of older soil carbon to $CO_2$ efflux, but differences in total $CO_2$ efflux responses to warming versus drying suggest two different mechanisms. Specifically, warming stimulated the decomposition of older soil carbon by increasing overall soil $CO_2$ efflux, with our results indicating a microbial switch in resource use following the depletion of fresh organic matter under warmed conditions[34]. In contrast, drying reduced total soil $CO_2$ efflux, apparently limiting the mobility of fresh carbon in soils and delivery to decomposers. This restriction of microbial access to fresh carbon explains the shift towards increased contributions of older carbon in total soil $CO_2$ emissions. Thus, climatic warming and drying will likely increase the vulnerability of previously stored soil carbon in tropical forests by stimulating the decomposition of old carbon.

## Results and discussion
### Effects of experimental warming on the average age of respired $CO_2$
We investigated the effects of soil warming at the Soil Warming Experiment in Lowland Tropical Rainforest (SWELTR)[15] during two seasonal timepoints. Soil warming increased the $\Delta^{14}C$ of respired $CO_2$ during the wet season, indicative of greater efflux of 'bomb' carbon under warmed and wet conditions (Fig. 2a). Specifically, the mean $\Delta^{14}C$

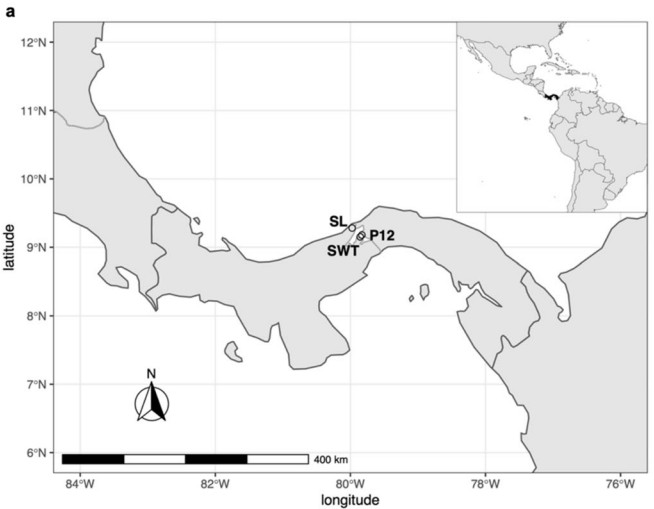
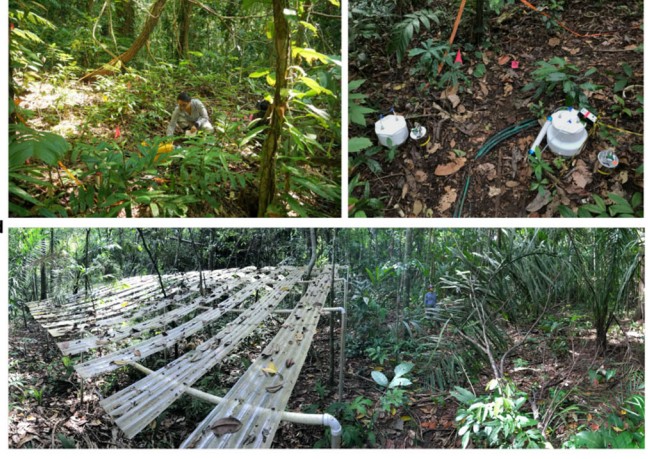

**Fig. 1 | Study site locations on the Isthmus of Panama. a** SWT = the Soil Warming Experiment in Lowland Tropical Rainforest (SWELTR) and SL = San Lorenzo. SL and P12 are Panama Rainforest Changes with Experimental Drying (PARCHED) sites. Map made with Natural Earth. Free vector and raster map data @ naturalearthdata.com. **b** Experimental plot at SWT. **c** Soil $CO_2$ efflux sampling for $^{14}C$ analysis with heating cables visible on the soil surface. **d** A pair of experimental plots at P12 where 50% throughfall exclusion structures are on the left and a paired control plot (with no throughfall exclusion) is shown on the right.

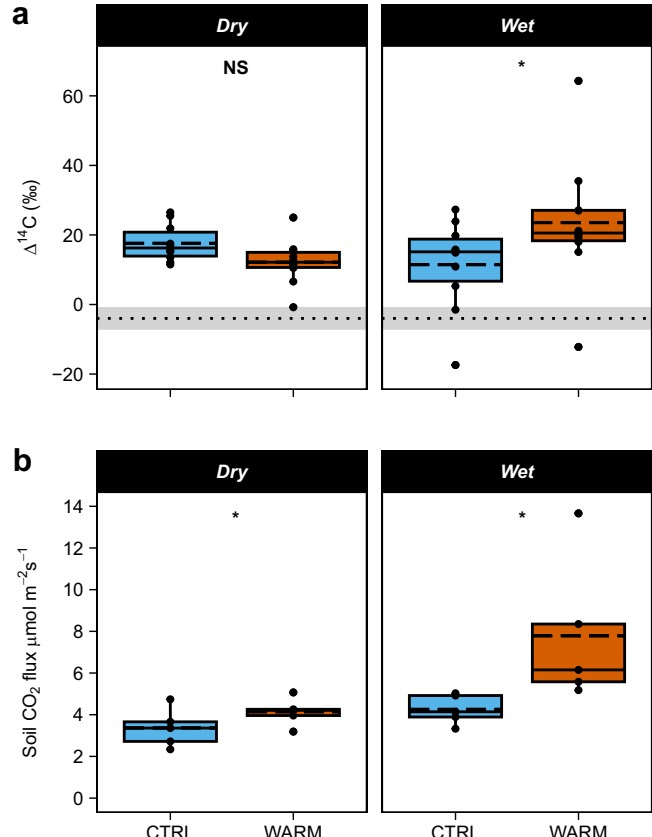

**Fig. 2 | Soil warming effects on $^{14}$C values and rates of soil $CO_2$ efflux during the dry and wet season in 2019. a** $^{14}$C of respired $CO_2$ from single time-point measurements for $n = 2$/plot (total $n = 10$). The mean $\Delta^{14}$C value of air samples collected from the sites in this study in 2019 was $-4 \pm 3$‰ and is indicated by the dotted reference line (mean) and gray shading (±standard error). Effects of experimental warming and season were tested using a three-way repeated measures ANOVA with collar type, treatment, and season (Supplementary Table 2). This test indicated a significant treatment by season interaction ($p = 0.02$), and a multiple comparisons test with a Holm adjustment indicated a significant effect of experimental warming in the wet season ($p = 0.03$) but not in the dry season ($p = 0.33$). **b** Monthly average total soil-respired $CO_2$ flux for March and October 2019 ($n = 5$ paired plots). Effects of experimental warming and season were tested using a two-way repeated measures ANOVA with treatment and season (Supplementary Table 3). This test indicated significant main effects for treatment ($p = 0.03$) and season ($p = 0.03$). The figures show plots warmed by +4 °C (WARM) and controls (CTRL). Solid lines indicate medians, dashed lines indicate means, ends of boxes show the upper (Q3) and lower (Q1) quartiles, whiskers indicate minimum and maximum ranges (calculated from quartiles), solid points are individual observations. Asterisks indicate statistically significant differences between control and warmed plots where $p \leq 0.05$ whereas NS indicates non-significant differences ($p > 0.05$).

of respired $CO_2$ was $12 \pm 5$‰ higher in warmed plots than control plots in the wet season ($p = 0.02$). In the warmed plots, $\Delta^{14}$C of respired $CO_2$ was also $11 \pm 5$‰ higher in the wet season than in the dry season ($p = 0.03$).

The observed increase in the $\Delta^{14}$C of respired $CO_2$ indicated that carbon fixed nearer to the bomb spike (circa 1963; see Supplementary Fig. 1 for reference), i.e., decadal-aged carbon, contributed more to soil $CO_2$ flux under warmer and wetter conditions (with warming in the wet season) compared to cooler and drier conditions (without warming and in the dry season). Because roots typically respire $CO_2$ with $\Delta^{14}$C values close to the current atmosphere[26,35], this result suggested that the shift was attributable to microbial (not live root) $CO_2$ flux. Thus, we found increased decomposition and loss of decadal-aged soil carbon under warmer and wetter conditions, while recently fixed carbon was

the dominant source of carbon respired under drier or cooler conditions.

During the time periods of our study, total soil $CO_2$ flux rates were also higher in warmed and wet conditions (Fig. 2b), with this difference in total flux likely driving increased utilization of older C. We found that total soil $CO_2$ flux increased from the dry season (March) to the wet season (October) by 60% ($p = 0.03$). Across seasons, experimental warming increased total soil $CO_2$ flux from $3.8 \pm 0.4$ μmol $CO_2$ m$^{-2}$ s$^{-1}$ in control plots to $6.0 \pm 1.4$ μmol $CO_2$ m$^{-2}$ s$^{-1}$ in warmed plots ($p = 0.03$). We partitioned the total soil $CO_2$ flux for the sampling periods in this study into heterotrophic (soil-derived) and autotrophic (root-derived) $CO_2$ flux using in situ root-exclusion and ingrowth cores (see section "Methods") and found that $74 \pm 7$ % of total soil $CO_2$ flux was heterotrophic (Supplementary Fig. 2). These results were consistent with a published 2-year time series of soil $CO_2$ flux from this experiment[15], which showed a 55% increase in soil $CO_2$ flux with warming attributed primarily to soil microbial (rather than live root) $CO_2$ flux.

Thus, our results strongly indicate that warming caused an increase in the emission of older carbon (with a higher $\Delta^{14}$C value) from soil organic matter (SOM) into the atmosphere, which can be explained by several mechanisms that are not mutually exclusive. First, warming-stimulated soil $CO_2$ efflux during the 18–24 months preceding our measurements[15] may have depleted the pool of fresh soil organic carbon (i.e., with a $\Delta^{14}$C value closer to 0‰) leading to a switch in microbial substrate use to older pools of carbon[36]. We found that bulk soils to 20 cm depth contained higher $\Delta^{14}$C than soil $CO_2$ efflux (Supplementary Fig. 3) and could have served as an older carbon source with a higher abundance of $^{14}$C. Large increases in microbial enzymatic activity and a shift in the microbial community composition in surface soils were detected at our study site after 2 years warming, which may have coincided with a shift in substrate availability and use[31]. Second, warming may have increased the degradation of older carbon pools via priming, whereby the rapid metabolism of plant-carbon inputs provided the necessary energy for microbes to synthesize enzymes to access longer lived, more chemically complex, carbon pools[34]. This process was observed in response to additions of fresh plant litter at a forest near our warming site[37]. In support of this mechanism, during the wet season in warmed soil we observed increased soil $CO_2$ efflux (Fig. 2b)[15], increased activity of soil extracellular enzymes[15,31], and increased variability in $\Delta^{14}$C of respired $CO_2$ (Fig. 2a), which suggested an increased connectivity of soil organisms to a wider variety of carbon sources available for microbial metabolism.

The idea that priming effects contributed to increased soil $CO_2$ emissions with warming was further supported by our finding that the largest increase in $CO_2$ efflux with warming occurred during the seasonal peak in leaf litter decomposition[15] (when high decomposition rates could result in priming of older soil carbon). Importantly, in laboratory incubations (without the carbon inputs that could drive priming under field conditions), the $\Delta^{14}$C values of microbial $CO_2$ efflux were similar for soil collected from warmed and control plots (Supplementary Fig. 3). Under field conditions, the SOM contributing to the increase in $CO_2$ efflux with warming likely originates from shallow soils (to about 20 cm depth). Shallow soils are disproportionately affected by plant C inputs via litter decomposition and fine roots. In addition, the high abundance of $^{14}$C in soils to 20 cm depth suggests that most of the carbon in the upper mineral soil layers is cycling on decadal timescales (Fig. S3). Together, these results suggest that, under field warming, microbes dwelling in the upper soil layers may have primed $^{14}$C-enriched SOM to increase the $\Delta^{14}$C in soil $CO_2$ efflux (Fig. 2a), or they might have depleted younger stocks of C and shifted substrate utilization. Further studies are needed to elucidate the specific mechanisms.

Other experiments that warmed the soil profile (by 4–4.5 °C) have reported increases in annual soil $CO_2$ flux of ~35% in temperate forest[14] and 14% in boreal peat forest[38], considerably lower than the 55%

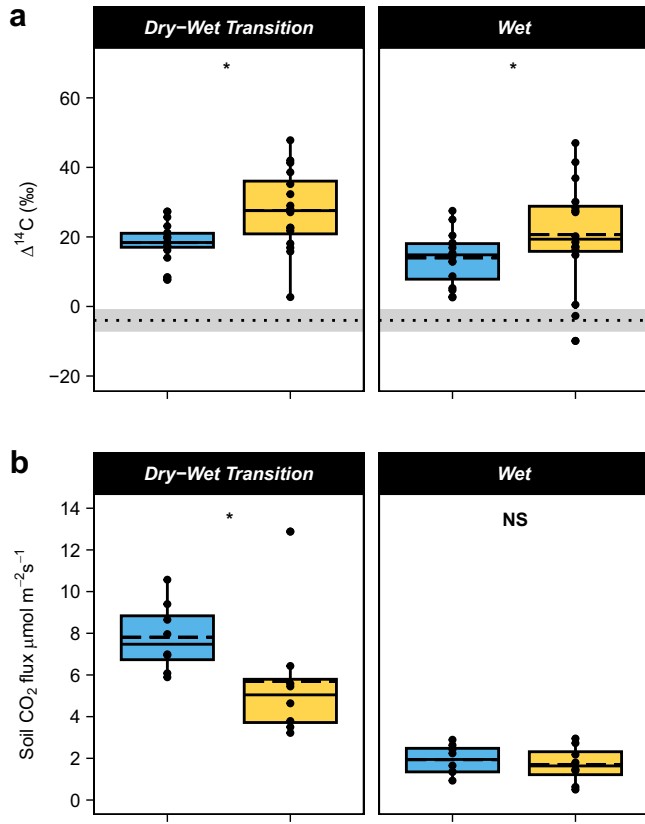

**Fig. 3 | Soil drying effects on $^{14}$C values and rates of soil $CO_2$ efflux during the dry-to-wet transition and wet season in 2019 from single time-point measurements at PARCHED. a** $^{14}$C of respired $CO_2$ for both P12 and SL for $n = 2$/plot (total $n = 16$). The mean $\Delta^{14}$C value of air samples collected from the sites in this study in 2019 was $-4 \pm 3$‰ and is indicated by the dotted reference line (mean) and gray shading (±standard error). Effects of throughfall exclusion and season were tested using a four-way repeated measures ANOVA with collar type, site, treatment, and season (Supplementary Table 5). This test indicated significant main effects for treatment ($p = 0.03$) and season ($p = 0.01$). **b** Single time-point total soil-respired $CO_2$ efflux for both P12 and SL ($n = 8$ paired plots). Effects of throughfall exclusion and season were tested using a three-way repeated measures ANOVA with site, treatment, and season (Supplementary Table 6). This test indicated a significant treatment by season interaction ($p = 0.05$) and a multiple comparisons test with a Holm adjustment indicated a significant effect of throughfall exclusion in the dry-to-wet season transition ($p = 0.02$) but not in the wet season ($p = 0.79$). The figures show plots with 50% of throughfall excluded (DRY) and controls (CTRL). Solid lines indicate medians, dashed lines indicate means, ends of boxes show the upper (Q3) and lower (Q1) quartiles, whiskers indicate minimum and maximum ranges (calculated from quartiles), solid points are individual observations. Asterisks indicate statistically significant differences between control and throughfall exclusion plots where $p \le 0.05$ whereas NS indicates non-significant differences ($p > 0.05$). Because $^{14}$C values and rates of soil $CO_2$ efflux did not differ between P12 and SL, the sites were pooled for statistical analysis. The sites are shown separately in Fig. 4.

## Effects of experimental drying on the average age of respired $CO_2$

We investigated the effects of ecosystem drying at two sites that are part of the Panama Rainforest Changes with Experimental Drying (PARCHED) study[33]. We selected the P12 site for its similarity and proximity to the warming experiment, with equivalent MAP and similar soils at the two sites. We also included the San Lorenzo (SL) PARCHED site, which is also on infertile soils but receives about 800 mm more annual rainfall than the other two sites, increasing the representativeness of our study for wetter tropical forests.

We found that experimental drying led to an increase in the mean $\Delta^{14}$C of respired $CO_2$ by $8 \pm 3$‰ averaged across sites and sampling periods ($p = 0.03$, Fig. 3a), consistent with a putative shift in microbial substrate use towards older, decadal-aged soil carbon. $CO_2$ $\Delta^{14}$C values also decreased by $6 \pm 3$‰ from the wet-to-dry season transition in May to the late wet season in November/December averaged across sites and treatment ($p < 0.01$). This is consistent with a depletion of the fresh carbon substrate delivered via dry season litterfall, over the course of the wet season, as supported by wet-season declines in total soil respiration and seasonal changes in soil biogeochemistry[11,33]. The $\Delta^{14}$C of respired $CO_2$ did not differ significantly between the drier (P12) and wetter (SL) sites (Fig. 4a, Supplementary Table 5), so sites were pooled for these analyses.

During our study period, experimental drying led to a 27% decrease in total soil $CO_2$ efflux, from $7.8 \pm 0.5 \, \mu mol \, CO_2 \, m^{-2} \, s^{-1}$ (control plots) to $5.7 \pm 1.0 \, \mu mol \, CO_2 \, m^{-2} \, s^{-1}$ (throughfall exclusion plots), during the dry-to-wet season transition for both sites ($p < 0.01$, Fig. 3b). This is consistent with longer term data over the first 5 years of this experiment, during which time these two sites had similar suppression of soil $CO_2$ flux in response to drying during the dry season and/or transitional seasons[11]. Soil $CO_2$ efflux partitioned into heterotrophic (soil-derived) and autotrophic (root-derived) components using field exclusion columns (see section "Methods"), suggested that this response was driven by a reduction in heterotrophic $CO_2$ flux rates ($p < 0.01$), with no apparent change in root-derived $CO_2$ efflux with drying (Supplementary Figs. 4 and 5). Overall, total, root, and heterotrophic soil $CO_2$ flux rates were higher during the dry-to-wet season transition compared to the wet season ($p < 0.01$), consistent with seasonal trends based on time series reported for these and other sites in the region[11].

As with warming, our observed patterns could be explained by several mechanisms that are not mutually exclusive. First, drying decreased soil respiration as moisture limited microbial activity and the transport of soluble carbon substrates from the forest floor into mineral soils[43] as observed following throughfall exclusion in other tropical forests[22,23,44]. Our finding that the $\Delta^{14}$C of respired $CO_2$ increased under partial throughfall exclusion (Fig. 3a) was consistent with reduced microbial access to fresh plant-carbon inputs. In our study, carbon in the 0–10 cm depth had higher $\Delta^{14}$C values than soil $CO_2$ efflux at both sites (Supplementary Fig. 6), suggesting that our observations may have been explained by a shift toward increased use of decadal-aged soil carbon (with a higher $\Delta^{14}$C value) by soil microbes. During the same period as our study, experimental drying caused a transition in the surface soil (0–10 cm depth) bacterial community composition toward a 'drought microbiome', which may have coincided with a microbial substrate shift. Indeed, other throughfall exclusion experiments reported decreased surface litter decomposition rates in Costa Rican forest[45], decreased $CO_2$ efflux from the litter layer in the eastern Amazon[46], and increased accumulation of forest floor material in temperate forest[29].

Second, our observed patterns could have resulted from decreased fine root production, respiration rates, or turnover with experimental drying. A change in root turnover or exudation could occur even in the absence of changing root respiration rates, as indicated by our data (no change in root respiration apparent in exclusion

increase in soil $CO_2$ flux with whole-profile warming reported for our site[15]. No change in the $\Delta^{14}$C values of respired $CO_2$ was reported by warming experiments in two temperate forests where emissions were sustained by decadal-aged C[14,39]. However, more similar to our $^{14}$C observations, soil warming increased $\Delta^{14}$C of respired $CO_2$ in a boreal forest, indicative of a greater contribution of decadal-aged carbon to the total flux[40]. Experimental warming also increased the age of $CO_2$ in porewater profiles in a boreal bog[41], in soil pore spaces in tundra[42], and in soil and ecosystem $CO_2$ flux in degraded permafrost[27]. These results suggest a variable, but potentially widespread, shift toward increased mobilization and loss of older soil carbon with climate warming.

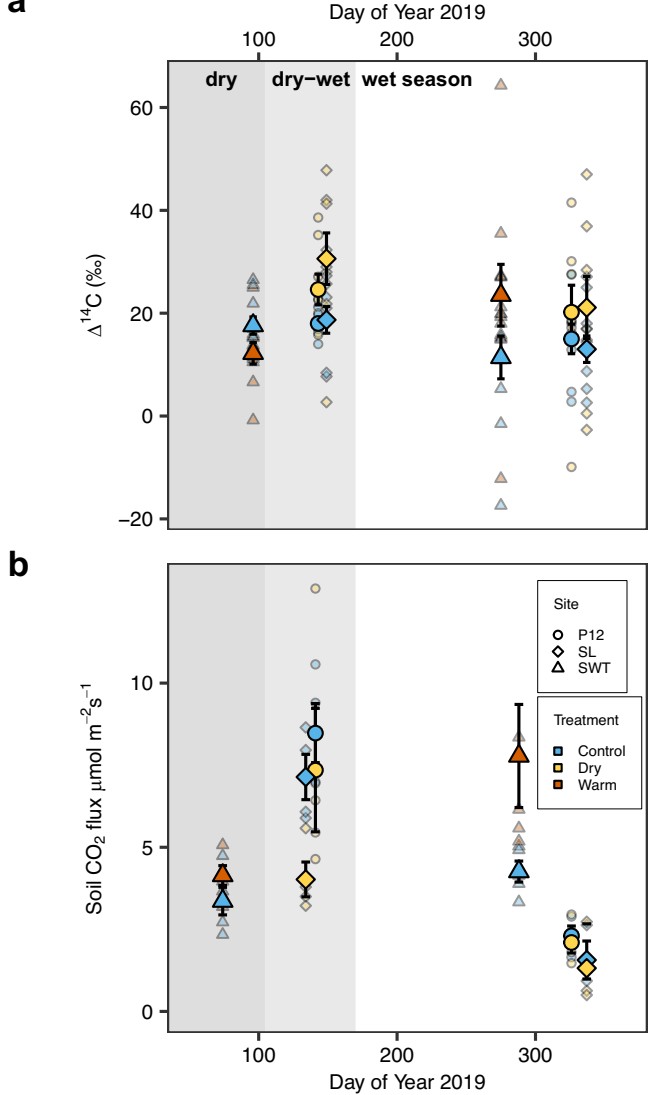

**a**

**b**

**Fig. 4 | Total soil-respired $CO_2$ flux and $^{14}C$ values over 2019 at SWELTR, P12, and San Lorenzo. a.** Single time-point $^{14}C$ of respired $CO_2$ at SWELTR ($n = 10$/treatment and time point) and at P12 and San Lorenzo ($n = 4$/site, treatment, and time point). **b** Total soil-respired $CO_2$ flux rates are monthly averages for SWELTR ($n = 5$/treatment and time point) and single time-point measurements at P12 and San Lorenzo ($n = 4$/site, treatment, and time point). The figures show means as large symbols with standard errors and individual measurements as small symbols. Dark gray shading denotes the dry season, light gray shading denotes the dry-to-wet seasonal transition period, and no shading denotes the wet season.

columns), and might explain why the increased $\Delta^{14}C$ of respired $CO_2$ with experimental drying during the dry-to-wet season transition persisted into the wet season, even as $CO_2$ efflux decreased. A shift in root growth toward greater depths to access available water is plausible as such a shift was observed in the Amazonian rainforest following experimental drying[23,47], and is a general pattern across tropical forests during dry seasons and droughts[48].

Third, our results could be explained by greater contributions of deeper, older soil carbon to surface soil $CO_2$ efflux as reported following throughfall exclusion in the eastern Amazon[23]. Deeper in the soil profile, soil carbon was older (Supplementary Fig. 6) and soil moisture was likely higher in dried plots[44], such that root and microbial activities were potentially less affected by changes in soil moisture. As a result, $CO_2$ produced at depth may have comprised a larger component of surface soil $CO_2$ efflux with experimental drying. At P12, soil

carbon pools to 50 cm depth that were enriched in $\Delta^{14}C$ relative to in situ surface $CO_2$ efflux and could have contributed to the increase in $\Delta^{14}C$ of respired $CO_2$ with seasonal or experimental drying, although older fractions of carbon in surface soils could also have contributed to this shift (Supplementary Fig. 6). At SL, however, subsurface particulate organic matter (originating mainly from roots) was depleted in $^{14}C$ relative to current atmosphere, indicating much older carbon, and could not have explained our observed increase in $\Delta^{14}C$ of surface $CO_2$ efflux (Supplementary Fig 6). Thus, an increase in the decomposition of deep soil carbon pools does not consistently explain the patterns we observed across both sites, and we conclude that a shift in substrate utilization toward older surface soil carbon, and/or changes in root turnover or decomposition, most likely explain our $^{14}C$ data.

### Seasonal effects on soil $^{14}C$ and $CO_2$ fluxes
The observed seasonal pattern in soil $CO_2$ efflux across sites (Fig. 4b) likely reflected a combination of favorable conditions for microbial activity during seasonal rewetting: litter accumulated over the dry season provided ample carbon substrate[20,49], dissolved organic carbon (DOC) production and transport facilitated microbial access to substrate[50], and rewetting of soil following drought strongly stimulated microbial activity[23,44,51]. Then, the effect of throughfall exclusion attenuated and the effect of soil warming became more pronounced during the later wet season as soils at all sites became more uniformly wet. In these seasonally moist forests, litter and particulate organic matter build up over the dry season and fuel the sharp increase in respiration rates that we report during the dry-to-wet season transition period at P12 and SL[11,20]. Indeed, studies in nearby forests including the warming site (SWT)[15] and elsewhere in the tropics[23,47] have shown a similar increase in soil respiration rates during this dry-to-wet transition period, driven by seasonal patterns of moisture availability, microbial biomass[33], and leaf-litterfall input[49].

### Potential combined effects of warming and drying
Climate change is expected to alter rates of soil carbon cycling in tropical forests through warming[52] and altered rainfall regimes[53] simultaneously. Unfortunately, no experiments have manipulated warming and drying together. Our results, based on individual responses to in situ experimental soil warming and soil drying in nearby forests, demonstrate that warming and drying not only change the emission of $CO_2$ from soils, but also the age of the carbon being emitted. Specifically, we found that both warming and drying increased the utilization of older soil carbon, even though warming increased but drying decreased soil $CO_2$ efflux. Thus, while further full factorial experiments are needed to elucidate the combined effects of warming and drying on soil carbon losses, our results indicate that warming and drying together will increase losses of older and previously stable soil carbon.

The consequences of our findings have wider implications when the impact of warming and drying on aboveground processes is considered alongside belowground effects. Both warming and drying have been shown to have detrimental effects on tropical forest productivity, based on in situ drying[53] and warming[54] experiments as well as observational studies during weather events[55]. Overall, these studies provide limited evidence for acclimation of photosynthetic activity. Thus, warming and drying together may decrease inputs of fresh carbon to soils, further exacerbating increased losses of older soil carbon observed here. Meanwhile increased $CO_2$ release from subsoil carbon is likely to continue under warming and drying, decreasing soil carbon storage throughout the soil profile, at least until drying extends to deeper soil horizons.

It is important to note that our reported increases in $^{14}C$ with experimental warming and drying reflected changes in the average age of carbon being respired (the equivalent of 2–3 years in the mean age of respired carbon, see Supplementary Fig. 1), but do not provide insight

into the distribution of carbon ages contributing to these averages. Our ongoing work using $\Delta^{14}C$ to study soil carbon storage and cycling in these and other sites along the Panama Isthmus rainfall gradient shows that soils in these forests store large proportions of young soil carbon compared with other ecosystem types, suggesting rapid turnover (Supplementary Figs. 3 and 6). Others have reported soil carbon turnover times of <10 years for numerous tropical forests[56,57]. While turnover times of decades to millennia were reported for clay-associated soil C at depth, the majority of soil C was found to have turnover times of less than a decade in Oxisols and Ultisols in Amazonian forest[58]. In forest in Puerto Rico, the mean carbon pool age for light density fractions (10–25% of the total C pool depending on depth) was 1–4 years to 60 cm depth[59], demonstrating the presence of an important and very rapidly cycling soil carbon pool even at depth. Thus, we conclude that our observed shifts in the age of respired carbon are striking, indicating a substantial increase in contributions from older soil carbon fractions, especially considering that these results were observed following relatively short-term (1–3 year) experimental treatments.

In summary, we demonstrate how warming and drying affect the rate and age of soil carbon emission to the atmosphere in tropical forests, by determining the $\Delta^{14}C$ of soil $CO_2$ efflux following experimental soil warming (whole-profile heating by 4 °C) and soil drying (50% throughfall exclusion). Experimental warming increased soil $CO_2$ efflux and, during the wet season, increased the age of respired soil carbon by roughly 2–3 years. In contrast, experimental drying decreased heterotrophic respiration rates, but also increased the age of respired soil carbon by roughly 2 years. Together, these results indicate an increase in the vulnerability of extant soil carbon, and a relative shift in microbial carbon use toward older sources: warming by depleting the pool of rapidly cycling carbon and stimulating the decomposition of old carbon; drying by reducing the mobility, accessibility, and subsequent decomposition of new carbon inputs. These findings imply a destabilization of old soil carbon under both warming and drying, which will have major implications for tropical forest–climate feedbacks. Our findings point to a need to study the effects of modified soil moisture and temperature together to capture and predict the net effects of climate change on tropical forest soil carbon storage now and in the future.

## Methods

### Study area and manipulation experiment descriptions

This study was conducted in three lowland tropical forest sites in central Panama. Two of the sites include experimental drying and one site includes experimental warming (Fig. 1 and Supplementary Table 1). Mean annual air temperature for all the sites is around 26 °C and air temperature is relatively constant over the year[60]. The region encompasses a precipitation gradient, with higher MAP to the north and lower MAP to the south, and has highly variable parent materials that influence available nutrients[61], but all three sites in this study are on low-fertility soils as described below.

SWELTR[15,31] is located on Barro Colorado Island in the middle of the precipitation gradient, receiving just under 2600 mm MAP. Soils at the site are moderately weathered, clay-rich, Dystric Eutrudepts (Inceptisols) formed on the conglomerate parent materials of the Bohio formation, primarily basalt and graywacke sandstone. The experiment includes 5 paired warmed and control plots (ten plots total). Soil warming started in November 2016 and is achieved using resistance cables buried to 1.2 m depth to warm the entire soil profile by an average of 4 °C above ambient temperature.

The two experimental dying sites included in this study consist of throughfall reduction experiments and are part of the PARCHED study[11,33]. The drier site (P12) is located on Buena Vista Peninsula, at 51 m above sea level, and receives a similar MAP to SWELTR. Like SWELTR, soils at P12 are low-fertility Ultisols formed on the Bohio formation. The wetter site (SL) is located closer to the Caribbean coast,

at 175 m above sea level, and receives about 3421 mm MAP. Soils at SL are low-fertility Oxisols formed on Chagres sandstone and contain more clay than soils at P12[33] contributing to overall higher soil moisture at SL than at P12. Each site includes four paired dry and control plots (eight plots total). SL has been previously referred to as Sherman Crane in the literature but has been renamed as the former evoked negative connotations associated with the legacy of colonialization. Throughfall reduction structures that exclude 50% of throughfall were installed over 10 × 10 m plots in June (P12) and July (SL) 2018 and remained fixed throughout the experimental period.

### Field sampling and data collection

At all three sites, each plot has replicate soil respiration collars (20 cm diameter at SWELTR and 10 cm diameter at PARCHED) with fluxes measured regularly with an LI-8100 infrared gas analyzer (LI-COR Biosciences) along with soil temperature and moisture. In addition, root-exclusion and root-ingrowth cores were installed for all three experimental sites (10 cm diameter, 30 cm deep PVC tubes), which are used to partition heterotrophic and autotrophic respiration flux corrected for disturbance following Nottingham et al.[15].

For SWELTR, we interpreted $^{14}C$ of $CO_2$ in the context monthly average total soil respiration rates, partitioned root and heterotrophic soil respiration rates, soil temperature, and soil moisture for March and October 2019. March was chosen rather than April because sampling for $^{14}C$ occurred the 1st week of April and early rains started later that month, marking the beginning of the transitional period from dry and wet season. Soil $CO_2$ efflux was measured biweekly using a Li-cor gas analyzer (IRGA Li-8100; LI-COR Biosciences), volumetric soil moisture at 0–10 cm depth was measured using a Thetaprobe (Delta-T sensor/Campbell) sensor and soil temperature at 0–10 cm depth was measured using an HI98509 thermometer probe (Hanna Instruments). To assess warming treatment effects on $^{14}C$ of bulk soil and $CO_2$ efflux in laboratory incubations, a soil core was collected from each plot in October 2019, after soil flux sampling, from the following depth increments: 0–10, 10–20, 20–50, and 50–100 cm. Soils were shipped immediately to Lawrence Livermore National Laboratory where they were processed and incubated in the laboratory. Soils were sieved to 2 mm and ground to a fine powder for $^{14}C$ measurement. Field-moist soils were picked free of large root fragments and incubated at field moisture at 26 °C until sufficiently high $CO_2$ concentrations were reached for $^{14}C$ analysis (4–7 days for 0–20 cm depths and 24–214 days for 20–100 cm depths).

For PARCHED, we interpreted $^{14}C$ of $CO_2$ in the context of instantaneous total soil $CO_2$ efflux, soil temperature, and soil moisture measured within 2.5 weeks of sampling for $^{14}C$–$CO_2$ (≤3 days except for SC in May due to logistical constraints). For all respiration rate sampling points, soil $CO_2$ efflux was measured using a Li-cor gas analyzer (IRGA Li-8100; LI-COR Biosciences), volumetric water content at 0–10 cm depth was measured using a ML-3 ThetaProbe read by an HH2 Moisture Meter, and soil temperature at 0–10 cm depth was measured using a digital external soil temperature 5-inch probe (Forestry Suppliers Part 89102). Soil cores were collected at P12 and SL prior to the construction of throughfall exclusion structures but near the experimental plots. Soils from the 0–10 cm depth were collected in 2018 and shipped immediately to Lawrence Livermore National Laboratory where they were processed and incubated in the laboratory as described above for the SWELTR soils. We also performed $^{14}C$ measurements on bulk soils and density fractionations from soils that were sampled in 2015 in the following depth increments: 0–10, 10–25, and 25–50 cm depths. Soils from 0–10 cm and 20–50 cm depths were density fractionated into dense, free light, and occluded light fractions using low C/N sodium polytungstate (SPT-0, Poly-Gee) adjusted to a density of 1.7 g cm$^{-3}$. A detailed description of the density fractionation method can be found in the Supplementary Methods.

We collected gas samples from soil chambers placed over soil respiration collars and root-exclusion cores, for the determination of

$^{13}$C and $^{14}$C in total soil respiration (soil + roots) and heterotrophic (root-free) soil respiration (we assumed that the $CO_2$ flux from the root-exclusion cores originated from SOM). Samples were collected from five sets of paired plots at SWELTR (ten plots total), from eight sets of paired plots at PARCHED (four control and four throughfall reduction plots at both P12 and SL), and during two sampling campaigns at each study site. At SWELTR, we collected samples during the dry season (April 6–7) and the wet season (October 2–3) of 2019, 2–3 years after the initiation of the warming treatment. At PARCHED, we collected samples during the dry–wet season transition (P12, May 22–24; SL, May 27–31) and the wet season (P12, November 21–23; SL, December 2–4) of 2019, roughly 1–1.5 years after the initiation of the drying treatment. Sampling occurred toward the end of each seasonal phase to represent when moisture limitation is at its lowest (late wet season) or greatest (late dry season). However, for PARCHED we sampled during the dry-to-wet season transition, which is when the difference in soil moisture between control and treatment plots is greatest–the dry season being effectively prolonged in treatment plots.

For gas collection, sampling collars were fitted with a static chamber lid constructed of PVC with two gas sampling ports. Chamber headspace was recirculated through a soda lime trap to scrub the headspace air of $CO_2$ for >4 times the chamber volume, using a battery-operated air pump. Chamber lid ports were closed to allow $CO_2$ to accumulate for long enough to ensure sufficient accumulation of $CO_2$ for $^{13}$C and $^{14}$C measurement (~1–4 h depending on flux rates). Headspace air was collected in an evacuated 1 L flask equipped with an 80 mL min$^{-1}$ flow restrictor to minimize isotopic fractionation during sampling. During each sampling campaign at each site, we also collected a reference air sample into a 3 L flask equipped with a 6 mL min$^{-1}$ flow restrictor.

### Radiocarbon interpretation
The ability to use $^{14}$C to indicate the C sources contributing to respiration stems from changes in atmospheric $^{14}CO_2$ over the last century from atmospheric thermonuclear weapons testing[62] (Supplementary Fig. 1). Atmospheric thermonuclear weapons testing in the late 1950s and early 1960s doubled the amount of $^{14}$C in the atmosphere. Following the atmospheric nuclear weapons test ban in 1963, the amount of $^{14}$C in the atmosphere decreased as this so-called bomb C was taken up by vegetation and oceans allowing annual to sub-annual resolution of the time of carbon assimilation from the atmosphere. Fossil fuel emissions contribute to the sustained decline of atmospheric $\Delta^{14}$C values over recent decades. Thus, each year a unique $\Delta^{14}$C signature is assimilated by plants via photosynthesis, translocated, and allocated to biomass growth and metabolism. The $\Delta^{14}$C of respired $CO_2$ can indicate an average age, or mean time elapsed since that carbon was fixed from the atmosphere, although it is important to recognize that this carbon is not homogenous in source or age. In the absence of extreme stress (e.g., girdling or complete defoliation), plants, including roots, tend to respire carbon from recent photosynthates, with $\Delta^{14}$C of respired $CO_2$ close to the atmosphere in that year[35,63]. In contrast, the $^{14}$C value of microbial respiration reflects the substrates the microbes are utilizing, which have an average age of several years or longer[26]. Studies have used the shift in the $\Delta^{14}$C value of soil-respired $CO_2$ to show changes in the source C pools supplying microbes over seasonal cycles[35] with disturbance including fire[64], warming[40], and drying[29]. Here, we interpreted shifts in the $\Delta^{14}$C value of soil-respired $CO_2$ with experimental warming or drying to reflect a shift in the average age of soil C substrates used for cellular metabolism and subsequent respiration as $CO_2$. For reference, we determined an approximate age shift with experimental warming and drying by fitting data to estimate the average year of synthesis (fixation from the atmosphere) using an annual decline in atmospheric $\Delta^{14}$C of 4.6‰ calculated from 1995 to 2019[62] (see Supplementary Fig. 1).

### $CO_2$ isotopic measurements
After collection, air samples were shipped to Lawrence Livermore National Laboratory's Center for Accelerator Mass Spectrometry where $CO_2$ from the field and laboratory incubation headspace was purified using cryogenic separation. For each air sample, a split of extracted $CO_2$ was analyzed for $^{13}$C at the Department of Geological Sciences Stable Isotope Laboratory at the University of California-Davis (GVI Optima Stable Isotope Ratio Mass Spectrometer). Each bulk soil and fraction sample was measured for $\delta^{13}$C at the Center for Stable Isotope Biogeochemistry at the University of California-Berkeley (Iso-Prime100 mass spectrometer) and was combusted to $CO_2$ in the presence of CuO for radiocarbon analysis. All samples were reduced to graphite and analyzed for $^{14}$C analysis on the Van de Graaff FN accelerator mass spectrometer. Measured $\delta^{13}$C values are reported relative to V-PDB and were used to correct $^{14}$C values for mass-dependent fractionation. $^{14}$C isotopic values are reported in $\Delta^{14}$C notation[65], had an average AMS precision of 3‰, and were corrected for $^{14}$C decay since 1950.

### Statistical analysis
Statistical analyses were performed in R v. 4.3.2[66] using two-, three-, or four-way analysis of variance (ANOVA) with repeated measures using the nlme (v. 3.1.164)[67] and lme4 (v. 1.1.35.1)[68] packages at $\alpha = 0.05$. Statistical results are in Supplementary Tables 2–9. Data were tested for normality and were not transformed. $\Delta^{14}$C values did not differ between total and root-free $CO_2$ flux (using root-exclusion columns; Supplementary Figs. 7, 8 and Supplementary Tables 2, 5), so root exclusion and total soil collars were pooled for tests of treatment and season effects. When present, interaction effects were investigated using the Phia (v. 0.3.1)[69] package with nlme or least squares means tests using the lsmeans (v. 2.30.0)[70] package with lme4 and with a Holm adjustment for multiple comparisons. Analyses were performed for SWELTR and PARCHED experiments separately and included effects of experimental treatment, season, and collar type (for isotopes only). The PARCHED ANOVA model included site to enable comparisons of the P12 and SL sites. In the text, results are reported as means followed by one standard error. Reported effect sizes are least squares means differences followed by one standard error.

### Reporting summary
Further information on research design is available in the Nature Portfolio Reporting Summary linked to this article.

## Data availability
The data generated and used in this study have been deposited at figshare [https://doi.org/10.6084/m9.figshare.24240211] and at the US Department of Energy's Environmental Systems Science Data Infrastructure for a Virtual Ecosystem (ESS-DIVE) [https://data.ess-dive.lbl.gov/datasets/doi:10.15485/2425968].

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

## Acknowledgements

We thank Maria Jose Montero for help with field sampling at SWELTR. We thank Makenna Brown, Biancolini Castro, Lily Colburn, and Korina Valencia for help with field sampling at PARCHED. We thank the Smithsonian Tropical Research Institute (STRI) for their assistance in ensuring that samples were collected and exported in a responsible manner and in accordance with relevant permits and local laws. These sites are active ecological research sites maintained by the Smithsonian Tropical Research Institute. Samples were collected and exported in a responsible manner. Any disturbance associated with accessing field sites and collecting samples was performed consistent with directives from STRI. All experimental work, sample collection, and sample export to the United States was done in compliance with local, national, and international laws and regulations with the necessary research, export, and import permits. This work was performed under the auspices of the U.S. Department of Energy by Lawrence Livermore National Laboratory under Contract DE-AC52-07NA27344 (LLNL-JRNL-853569). This work was funded by the Office of Biological and Environmental Research in the U.S. Department of Energy Office of Science through award SCW1572 to K.J.M. and DE-SC0015898 to D.F.C. The study was further supported by a UK NERC Grant NE/T012226 to A.T.N. For the purpose of open access, the author has applied a Creative Commons Attribution (CC BY) license to any author accepted manuscript version arising from this submission.

## Author contributions

K.J.M.: performed radiocarbon and statistical analyses, performed gas sampling and $CO_2$ purification, designed and funded the radiocarbon experiment with input from D.F.C. and A.T.N., performed data analysis, and wrote the manuscript. D.F.C.: provided data, led PARCHED, collected soil samples, helped write and edit the manuscript. L.H.D.: provided data, performed gas sampling, led field operations for PARCHED, collected soil samples, helped write and edit the manuscript. A.L.H.: prepared for and performed gas sampling at all study sites and helped edit the manuscript. K.M.F.: performed laboratory incubations and density fractionation, prepared samples for isotopic and elemental analysis, and helped write and edit the manuscript. A.T.N.: provided data, led SWELTR, collected soil samples, and helped write and edit the manuscript.

## Competing interests

The authors declare no competing interests.
