## [Peer Review File · Nature Communications]

Experimental Warming and Drying Increase Older Carbon Contributions to Soil Respiration in Lowland Tropical ForestsREVIEWER COMMENTS

Reviewer #1 (Remarks to the Author):

This study found that warming led to an increase, while drying resulted in a decrease in soil respiration within a lowland tropical forest. The ^{14}C abundance in the carbon flux was higher in the warmed plot and the dry plots. The authors concluded, based on the ^{14}C data, that microbes utilized more older carbon under warming conditions and less fresh carbon under drying conditions. While this contributes interesting insights to the existing literature on investigating carbon sources under climate change, several significant concerns undermine the validity of the conclusion.

The claimed age of soil carbon flux, as indicated in the title, was not quantified in the study. Given that age was neither quantified nor described in the results, it may be prudent to reconsider its inclusion in the title. Additionally, the study revealed a 2-3 year age difference, which appears negligible when considering the overall age of soil carbon (thousands of years).

The absence of crucial data for tracking sources and substantiating the conclusions is notable. Soil radiocarbon content across different depths is essential to unveil the contribution of deeper soils to the carbon flux, and the inclusion of radiocarbon data from litter and roots is equally significant. It would enhance the study's value if the authors could quantify the contribution of various soil layers to the overall carbon flux under warming and drying conditions, similar to the approach in the reference (DOI: 10.1111/gcb.12058).

Contrary to the assertion in the title, the study found that warming did not impact the ^{14}C levels during the dry season. This observation contradicts the initial claim made in the title.

The conclusion drawn appears to lack robustness, primarily due to the reliance on a one-year sampling period for radiocarbon, as indicated by its considerable variation (Figure 1a, 2a). While the responses of soil respiration to warming or drying appear to be consistent, the reaction of radiocarbon is relatively minor and accompanied by substantial uncertainty. This suggests that the conclusion may be subject to alteration in different years.

Previous findings from others are very relevant and needs to be incorporated in the introduction.
Example list:

1. Thawing permafrost increases old soil and autotrophic respiration in tundra: Partitioning ecosystem respiration using $\delta^{13}\text{C}$ and $\Delta^{14}\text{C}$
2. Old soil carbon losses increase with ecosystem respiration in experimentally thawed tundra
3. Five years of whole-soil warming led to loss of subsoil carbon stocks and increased CO_2 efflux
4. ^{14}C evidence that millennial and fast-cycling soil carbon are equally sensitive to warming

Some more comments

L70-72: Judging by figure 1a (the two horizontal lines in the boxplot), the difference in ^{14}C does not

look greater than 10 per mil between control and warmed plots in wet season. And the difference between dry and wet season looks greater than the difference between control and warmed plots. Please double check.

L77-79: if the logic works, should not ^{14}C in total flux be lower than heterotrophic respiration?

L116: please see the reference list above for other publications and Edward Schuur's group publication.

L169-172: soil radiocarbon content along depth is key

Reviewer #2 (Remarks to the Author):

Thank you for the opportunity to review this interesting manuscript. In this paper, "Experimental Warming and Drying Increase the Age of Soil Respired Carbon in Lowland Tropical Forests", the authors present data from a soil warming and drought experiment on low-fertility tropical rainforest soils. The data show that warming leads to a loss of older carbon, and more carbon over all, while drying decreased decomposition of recent inputs. These data are presented clearly and the interpretations are well-supported without being speculatively. Generally, this is an excellent manuscript that should require little editing before publication.

The main questions/issues/ideas that persist are

- 1) Why was the soil warming $+4^\circ\text{C}$?
- 2) How were radiocarbon data interpreted? Was this simply by fitting data to estimate year of synthesis? Or by using a one-pool model? These provide slightly different answers, with the latter being more reliable. Please explain in the methods.
- 3) Perhaps there could be a brief discussion of how the loss of old C and accumulation of younger plant-derived C shifts the overall SOM pool towards less stabilized C that is even more vulnerable to decomposition. This is mentioned in the abstract and conclusion but not in the main text.

Below are line-by-line comments, which are mostly minor.

L167-169: This sentence is a bit misleading. I believe the intention is "... the turnover of older decadal-aged carbon was less affected by drying than *was* the turnover of recent plant carbon inputs." The way it is written, it seems that the turnover of recent inputs more strongly affected the turnover of older C. Another approach could be "... the turnover of recent plant inputs was more strongly affected than older C to drying."

L214: If the warming increases C loss (that is older) but drying reduces respiration while preserving younger C (through reduced utilization of inputs), then indeed the combined effects would probably be that C lost is older. This is addressed in the next paragraph, but my initial read of this sentence led me to question the result. Perhaps this can be slightly re-ordered to keep the reader from questioning the reasoning, possibly just by re-assessing the word "thus".

L220: This sentence was slightly confusing due to splitting the two effects. Consider adding “both”: “... warming and drying together may decrease both inputs of fresh carbon to soils as well as degradation rates...”

Reviewer #3 (Remarks to the Author):

The authors demonstrated that soil carbon in tropical forest ecosystems is vulnerable to increased losses from heating and drying. The pathway to losses from heating centered on destabilization of decadal cycling soil C pools, while drying appeared to decrease stabilization of recent soil C inputs. The net impact of these two mechanisms indicate that the combination of heating and drying is likely to lead to a decrease in soil C sink strength for tropical forests. The novel approach taken in this study was to combine data from both heating and drying experiments in situ, relying on the power of radiocarbon measurements of respired CO₂. Using the bomb curve to inform the interpretation of the data points clearly to a decadal cycling pool as a prime contributor to soil C losses, while pairing the radiocarbon data with measurements of CO₂ production permits identification of mechanistic differences driving these responses under heating versus drying conditions. The authors effectively use the published literature along with their data to test mechanistic hypotheses, resulting in an important contribution to the field.

Radiocarbon is a powerful tool, but interpretation of the data can be challenging. Putting the $\Delta^{14}\text{C}$ values observed in the context of the atmosphere for the year of sampling would be a useful point of reference. For example, some of the observations of total soil $\Delta^{14}\text{C}$ respired during the wet season at the SWELTR site are less than 0‰ in both the control and treatment sites, indicating that priming of centennially cycling soil C may be occurring during the wet season regardless of treatment. The inclusion of the atmospheric $\Delta^{14}\text{C}$ record for the bomb-C period in the supplemental information is helpful in this regard, but additionally mentioning the $\Delta^{14}\text{C}$ of the current atmosphere in the main text would be helpful. For example, in lines 102:105 the authors suggest that warming during the initial phase of the SWELTR experiment may have depleted more recent soil C inputs in the warmed plots but not the control plots. However, the $\Delta^{14}\text{C}$ measurements made during the dry season are closer to the atmosphere in the warmed plots than the control plots, suggesting potentially greater contributions of recently fixed soil C in the treated plots (although the differences are not statistically significant).

Inferring exact ages from radiocarbon observations alone requires a model, as soil is an open and heterogeneous system, i.e., continually receiving new C inputs while simultaneously producing CO₂ emissions (Trumbore, 2000, “Age of soil organic matter and soil respiration: radiocarbon constraints on belowground C dynamics”). Although a model is required to ascribe ages, observations of relatively enriched $\Delta^{14}\text{C}$ values in soil C emissions from treated versus control plots is a clear indication that the primary source of emissions in the treated plots is C that was fixed after the bomb-C spike. The authors do acknowledge that soil C pools with a range of ages may be contributing to the observed fluxes, which is an important caveat for this analysis.

Estimating differences in soil C transit times in response to treatment would be one approach for obtaining a more quantitative estimate of the treatment effect on soil C cycling rates; this could be accomplished using a simple SoilR type model (Sierra et al., 2014, “Modeling radiocarbon

dynamics in soils: SOILR version 1.1”; Stoner et al., 2021, “Soil organic matter turnover rates increase to match increased inputs in grazed grasslands”). Such a model could also be used as a conceptual tool to test the hypotheses advanced here, even if the data were not available for adequately constraining soil C pool sizes and turnover rates (Beem-Miller et al., 2021, “Impacts of drying and rewetting on the radiocarbon signature of respired CO₂ and implications for incubating archived soils”). For example, it would be interesting to assess whether there is a difference in the effect on transit times for drying versus heating. Such a difference seems possible given that a) $\Delta^{14}\text{C}$ of respired CO₂ in the dry treatment plots of the PARCHED site was more enriched than that observed in the warmed plots of the SWELTR site, and b) the difference in treatment effects on CO₂ fluxes between the two experiments. Demonstrating differences in transit times are a clear way to estimate future soil C storage (cf. Sierra and Crow, 2022, “The climate benefit of sequestration in soils for warming mitigation”). However, the data are clear as is, and admittedly, such a modeling exercise would introduce new uncertainties, so may be outside the scope of the present work. Heterogeneity of soils is a major challenge for in situ warming experiments (and indeed, in soil organic matter science generally). The fact that the results in this study were robust in spite of this heterogeneity are promising. The authors point to the increase in heterogeneity observed during the wet season at SWELTR as potential evidence for priming of older soil organic matter decomposition alongside more recent soil C inputs, which is compelling when set alongside the references to existing literature on the importance of priming in nutrient limited systems such as these, and the observed increase in CO₂ fluxes. Vertical gradients in soil conditions are another key aspect of heterogeneity, and the authors posit that the decreasing impact of soil moisture with depth could be driving an increase in the relative contribution of respired CO₂ from deeper soil layers in the dry plots of the PARCHED experiment. This seems plausible given the decrease in respiration and the increase in the apparent age of respired C.

The authors pay special attention to seasonality, which is a major driver of soil CO₂ emissions in these ecosystems. Focusing on the time of the year with the greatest expected treatment effect clearly proved effective for testing the mechanistic hypotheses advanced in this study. However, it would have been helpful to put the data from this study in the context of the annual losses due to treatment observed for the sites, e.g., referencing the numbers provided by Nottingham et al., 2020 (“Soil carbon loss by experimental warming in a tropical forest”) for SWELTR, and given the decrease in respiration observed in response to the drying treatment, an annual estimate for PARCHED, e.g., Cusack et al., 2023, “Soil Respiration Responses to Throughfall Exclusion Are Decoupled From Changes in Soil Moisture for Four Tropical Forests, Suggesting Processes for Ecosystem Models”. This seems to be a missed opportunity for comparing the relative impacts of warming and drying (and the potential combined impact), which as the authors point out, will be critical for improving our understanding of the impacts of climate change on soil C cycling in tropical forest ecosystems.

Additional notes:

Fig. S2: “dry” panel mislabeled as “NA” and panels are missing letters.

Line 174: But what is the mechanism for which the older C would remain accessible, while new C inputs are not? Increased contributions from deeper soil C, which are less affected by soil moisture, seems more plausible.

NCOMMS-23-50060-T

Experimental Warming and Drying Increase Older Carbon Contributions to Soil Respiration in Lowland Tropical Forests

Our responses to the reviewers' helpful comments are provided below (in italics font), with changes tracked in the document and line numbers referencing the revised draft.

Yours Sincerely,
Karis McFarlane on behalf of all co-authors

Reviewer #1 (Remarks to the Author):

This study found that warming led to an increase, while drying resulted in a decrease in soil respiration within a lowland tropical forest. The ^{14}C abundance in the carbon flux was higher in the warmed plot and the dry plots. The authors concluded, based on the ^{14}C data, that microbes utilized more older carbon under warming conditions and less fresh carbon under drying conditions. While this contributes interesting insights to the existing literature on investigating carbon sources under climate change, several significant concerns undermine the validity of the conclusion.

The claimed age of soil carbon flux, as indicated in the title, was not quantified in the study. Given that age was neither quantified nor described in the results, it may be prudent to reconsider its inclusion in the title. Additionally, the study revealed a 2-3 year age difference, which appears negligible when considering the overall age of soil carbon (thousands of years).

Author response:

We appreciate the reviewer's suggestion regarding the use of "age" in the title and have revised the title as follows: "Experimental Warming and Drying Increase Older Carbon Contributions to Soil Respiration in Lowland Tropical Forests". We initially felt that the use of "age" was more approachable for the broad range in scientific expertise held by the readers of Nature Communications, but we appreciate the concern of the use of "age" in this way. "Decadal-aged" is more accurate to our findings but may not be clear in terms of meaning to readers unfamiliar with ^{14}C data used in the context of carbon cycling. Thus, we have used "older" as we only describe the difference in ages in manuscript.

However, we disagree with the second point because in the context of tropical forest carbon cycling where carbon is cycled extremely rapidly (e.g. plant-litter inputs decomposed within weeks), a difference of 2-3 years is indeed significant (in contrast to permafrost or peat soils, which cycle C more slowly). Furthermore, the average age of carbon respired from soil is younger than the average age of the soil pools the flux is sourced, in general. As such, these responses, after only 3 years of experimental warming

and 1 year of experimental drying, are notable because they reflect a rapid response to climatological change that could become increasingly significant for the overall C sink strength of tropical forest soils in the future. Finally, we note that our study was not designed to test the response of extremely old C (>1000 years) to warming; at our sites (See in the new Supplemental Figures 3 and 6 that we only see C ages > 1000 years in bulk soils below 50 cm). Rather, our approach quantified the release of decadal (>50 y) age C pool (implicitly represented by ¹⁴C bomb-labelled C), which represents an important C pool. We emphasize these points in our discussion (see Lines 482-506; and see title change).

Reviewer 1:

The absence of crucial data for tracking sources and substantiating the conclusions is notable. Soil radiocarbon content across different depths is essential to unveil the contribution of deeper soils to the carbon flux, and the inclusion of radiocarbon data from litter and roots is equally significant. It would enhance the study's value if the authors could quantify the contribution of various soil layers to the overall carbon flux under warming and drying conditions, similar to the approach in the reference (DOI: 10.1111/gcb.12058).

While we agree with the reviewer that clarification of the source and mechanisms behind our observed treatment effects would be very compelling, we strongly assert that the data in our present study provides important and novel insights into the effects of soil warming and drying on the tropical soil C cycle. We have added additional unpublished data that were intended for different manuscripts to add to our discussion. We thank the reviewer for their suggestions, some of which we are already pursuing in a follow up study (pending further grant awards to enable additional sampling and analyses).

Regarding the specific suggestions, soil ¹⁴C depth profiles at SWT (collected in 2019 from all treatment plots) and from P12 and SL (collected outside of the treatment plots before throughfall exclusion structures) are now included in the new Supplementary Figures 3 and 6 and discussed in Lines 218-232 and Lines 391-428. These data show that increased $\Delta^{14}\text{C}$ values in surface soil CO_2 efflux with warming or drying could be explained by increased decomposition of surface occluded light fraction, surface dense fraction, or subsurface light fraction. The reviewer also requests root and litter ¹⁴C data. Cordiero et al., in review include ¹⁴C values for roots at P12 and SC, which we refer to in the revision (Lines 389-391) as they find deep roots' ¹⁴C values range widely and decomposition of these roots could theoretically contribute to increased ¹⁴C values with drying (or warming). We have assumed that litter, which accumulates during the dry season and then decomposes during the wet season, shallow roots, and root respiration have ¹⁴C values similar to current atmosphere but will validate this in the future with litter measurements and root incubations.

The paper the reviewer cites (Hicks Pries et al., 2012) used laboratory incubations of degraded permafrost soils to identify the ¹³C and ¹⁴C end members of shallow soil and deep soil and refer to shallow soil as "young soil" and deep soil as "old soil". This approach is more feasible in the case of permafrost (or peat) where soil and carbon accumulate in

distinct layers above older soil layers, but it is problematic in the case of tropical forest soils where the accumulation of carbon in situ coincides with mineral weathering, high rates of vertical flushing, and young C enters deep soil through DOC leaching as well as root turnover and exudates. We performed laboratory incubations on soils collected from the SWT warming and control plots in 2019 but found no treatment effect on the $\Delta^{14}\text{C}$ values of respired CO_2 during incubation – possibly because soils are isolated from plant C inputs in the laboratory (see Lines 218-232 and Supplementary Fig. 3). Importantly, we point out that sampling, transporting, and processing of soil samples for laboratory incubations includes disturbance to soil structure, which can be important in protecting organic C from microbial access in soil aggregates or anaerobic microsites – this disturbance may contribute to differences between laboratory and in situ observations observed elsewhere (Phillips et al. 2013). We also performed pre-treatment incubations on surface soils from P12 and P13, which are now shown in Supplementary Fig. 6 and discussed in Lines 391-428, but in 2019 thought that one year of experimental drying would not be long enough to alter bulk soil C. In the future, now that we have regained site access after the COVID pandemic, we will consider incubations of soil from multiple depths, which could indicate relative respiration rates (albeit the use of incubations to relate to in situ fluxes is quite problematic) and isotopic values for heterotrophic respiration under experimental warming and drying – especially if the installation of gas wells is not feasible under funding restraints as this would best be addressed with measurements of the concentration and ^{14}C of CO_2 from multiple soil depths in situ as has been done elsewhere (Hicks Pries et al. 2017; Phillips et al. 2013).

In summary, we thank the reviewer for their suggestions for further measurements. The data presented in this paper and described in this response are informing and serving as baseline data for future work when changes to soil carbon pools are more likely to be detectable. At that time, we will be sure to address the points raised by the reviewer regarding validating assumptions related to root and litter isotopic values and address the potential contributions of shifts in CO_2 production at depth with profile gas measurements and/or incubations. We have added 2 new supplementary figures with the data we can provide and discussion of these data (Supplementary Figures 3 and 6 and Lines 218-232 and Lines 391-428). However, we strongly assert that our current study and the data we report is interesting and offers extremely novel insights on the impact of warming and drying on surface emissions of CO_2 .

Reviewer 1:

Contrary to the assertion in the title, the study found that warming did not impact the ^{14}C levels during the dry season. This observation contradicts the initial claim made in the title.

We see the reviewer's point but suggest that the overall effects of warming and drying can be summarized as increasing the ^{14}C value of respired CO_2 on an annual basis, especially considering that CO_2 fluxes at the warming experiment site were higher in the wet season, when there was an increase in the ^{14}C of respired CO_2 , than in the dry season when there was no treatment effect on ^{14}C . Our revised title is more tightly aligned to our experimental

findings, and we hope that this revised title is now more satisfactory for the reviewer.

Reviewer 1:

The conclusion drawn appears to lack robustness, primarily due to the reliance on a one-year sampling period for radiocarbon, as indicated by its considerable variation (Figure 1a, 2a). While the responses of soil respiration to warming or drying appear to be consistent, the reaction of radiocarbon is relatively minor and accompanied by substantial uncertainty. This suggests that the conclusion may be subject to alteration in different years.

Author response:

We appreciate the reviewers concern and agree entirely that more data would be useful to further reduce the uncertainty on this response. However, it is very important here to note that our results are based on well-designed manipulation experiments with replicated paired treatment and control plots. Showing a statistical difference between these plots offers robust experimental evidence for an effect for the specific sampling month. It would be interesting to perform time-series of measurements to obtain further detail on the seasonality of these patterns, which we hope to conduct in the future pending further funding acquisition. We further point to accompanying papers that show the full seasonality of CO₂ emission at high temporal resolution (Nottingham et al. 2020; Cusack, Dietterich, and Sulman 2023), which are consistent with the CO₂ emission in the present study – thereby strengthening our findings and allowing us to infer that the ¹⁴C-CO₂ data reported here are representative.

We assert that our findings based on the reported data are strong, given our robust experimental design; and especially when accompanied by other treatment effects reported for these experiments during the same or similar duration. We have increased references to these papers in the Results and Discussion. Over this same experimental period, warming treatments increased total soil CO₂ efflux rates by 55% and flux partitioning indicated this response was driven primarily by soil microbes (Nottingham et al. 2020). After just 2 years of warming, the soil microbial community shifted towards a less diverse, “thermophilic microbiome” and enzyme activities increased (Nottingham et al. 2022). Similarly, during this same experimental period, our team found that: 1) experimental drying decreased total soil CO₂ efflux by about 20% relative to controls during some seasons at P12 and SL) (Cusack, Dietterich, and Sulman 2023); 2) experimental drying decreased soil microbial biomass carbon and water extractable soil carbon, potentially decreasing dissolved organic carbon inputs to soils (Dietterich et al. 2022); and 3) experimental drying shifted soil microbial communities to a “drought microbiome,” such that throughfall-exclusion plots were enriched in actinobacteria relative to controls at P12 and SL (and another low fertility site) within 12–18 months (Chacon et al. 2023). These findings suggested that even in this relatively short time, we might see the shifts in the carbon sources contributing to soil CO₂ fluxes as we report in this study.

Here, we present short-term responses to experimental warming (~ 3 years) and drying (<2 years), as has been presented previously for early results from other manipulation experiments based on a limited sampling period (Hicks Pries et al. 2016; Hicks Pries et al. 2017; Wilson et al. 2016). We very much hope to be able to repeat this effort, provide a longer time series of ¹⁴C measurements, and more fully address the ambiguities regarding timescales of responses as well as the sources of C contributing to respiration in general and in response to warming and drying treatments – similar to how other manipulation experiments have followed up early, short duration responses to treatment with studies reporting longer term responses that benefit from additional data streams and are better able to identify mechanisms behind treatment responses (Soong et al. 2021; Wilson et al. 2021; Melillo et al. 2017; Schuur et al. 2023). The studies referenced here point to the importance of long-term experiments, but also to the relevance of short-term treatment responses in understanding the impacts of abrupt shifts in environmental conditions. Our experiments, though impacted by the global COVID pandemic, are ongoing, and we are actively pursuing opportunities to follow our early findings, including those reported here, with longer-term data.

We addressed this in the text with added references to and discussion of other results from our experiments over the same or similar study duration, to clarify this point (Lines 75, 192-222, 315-318, 338-380)

Reviewer 1:

Previous findings from others are very relevant and needs to be incorporated in the introduction. Example list:

1. Thawing permafrost increases old soil and autotrophic respiration in tundra: Partitioning ecosystem respiration using $\delta^{13}\text{C}$ and $\Delta^{14}\text{C}$
2. Old soil carbon losses increase with ecosystem respiration in experimentally thawed tundra
3. Five years of whole-soil warming led to loss of subsoil carbon stocks and increased CO₂ efflux
4. ¹⁴C evidence that millennial and fast-cycling soil carbon are equally sensitive to warming

Author response:

We thank the reviewer for the suggestion that we include more of the relevant literature in the Introduction. We have expanded the Introduction paragraph introducing the use of ¹⁴C to reveal the sources of C contributing to shifts in soil CO₂ efflux (Lines 46-67). In doing so, we have added several references, including suggestion #3 (Soong et al. 2021). However, we are limited in the extent to which we can provide a full literature review on the research using ¹⁴C in the context of soil respiration and warming as Nature Communications limits references to 70 and we would be remiss not to include in this paragraph some references to drying experiments as well as warming experiments. As such, we have had to make some difficult decisions here and throughout our paper regarding which papers to cite, have prioritized those most relevant to our study, and have opted to cite one paper that

covers several aspects of the same experiment or related work. [Note that additional studies are referenced in Lines 236-247 and 644-650.

Reviewer 1:

L70-72: Judging by figure 1a (the two horizontal lines in the boxplot), the difference in $\delta^{14}\text{C}$ does not look greater than 10 per mil between control and warmed plots in wet season. And the difference between dry and wet season looks greater than the difference between control and warmed plots. Please double check.

Author response:

We apologize for the misunderstanding here, which arose by mistaking the lines in our box plots as means, whereas they represent medians; this is further clarified in the revision. We have added a dashed line to the box plots that indicates the mean in Figures 2 and 3 in the revision and describe the line markings in the captions. As in the earlier draft, means and standard errors are also shown in the scatter plot (Figure 4 in the revision). The difference reported in the text is also clarified on Lines 129 and 257. It is also stated in the methods that values reported in the text are means followed by one standard error (Line 690).

To address this, we added dashed lines to indicate medians in the boxplot figures (now Fig. 2 and 3) and specify in the boxplot figure captions that the solid lines indicate the medians and dashed lines indicate the means.

Reviewer 1:

L77-79: if the logic works, should not $\delta^{14}\text{C}$ in total flux be lower than heterotrophic respiration?

Author response:

Yes, this is what one would expect. We suspect that we did not observe a difference between collar types because of the relatively low contribution of root respiration to the total soil CO_2 flux – as stated in Line 189 of the revision, “74 ± 7 % of total soil CO_2 flux was heterotrophic”. We can demonstrate using a mixing model that this partitioning of total soil respiration into heterotrophic (74%) and autotrophic (26%) components, the average $\delta^{14}\text{C}$ value from the total soil flux collars (19‰), and average $\delta^{14}\text{C}$ value from the root-free flux collars (14‰) would provide a $\Delta^{14}\text{C}$ value for autotrophic respiration of -2 to + 1 ‰. This is consistent with root respiration $\Delta^{14}\text{C}$ values near 2019 atmospheric values, both values expected based on the atmospheric curve and the air samples we collected at our sites in 2019, which were -4 ± 8 ‰ as shown by the new reference line in Fig. 1a and elsewhere. As we rely on the field exclusion collars for partitioning respiration flux, we did not measure the isotopic endmembers of root respiration using incubations. We will do this in the future to verify our assumptions that root respiration will have a similar $\Delta^{14}\text{C}$ value to the atmosphere and thank the reviewer for the suggestion.

Considering the potential for this statement to confuse our message, we have moved it, and the similar statement for the drying experiment, to the Methods section (Lines 675-678).

Reviewer 1:

L116: please see the reference list above for other publications and Edward Schuur's group publication.

Author response:

Thank you for the suggested references, in the revision we have been careful to include those we consider most relevant to our study. Considering each of them respectively, the Vogel et al., 2014 paper was cited in the original manuscript, remains in the revision, is from Ted Schuur's group, and is a field warming experiment. Hicks Pries et al., 2013 looked at permafrost thaw over the growing season in an area experiencing increasing thaw associated with climate warming, it does not include an experimental warming treatment and so we have not elected to cite it considering the limitation to 70 references. The reviewer is correct that we omitted the CIPHER permafrost warming experiment, which uses snow fences to increase soil temperatures and increase the thaw layer depth. We have added it to Line 248 but using the most recent reference (Schuur et al. 2023), rather than Hicks Pries et al., 2016. Note this reference is also cited in the new Introduction paragraph (Line 57-59).

Reviewer 1:

L169-172: soil radiocarbon content along depth is key

Author response:

As stated above in detail, we have added these data to the new Supplementary Figures 3 and 6 and discuss these and other new data in Lines 218-232 and Lines 391-428.

Reviewer #2 (Remarks to the Author):

Thank you for the opportunity to review this interesting manuscript. In this paper, "Experimental Warming and Drying Increase the Age of Soil Respired Carbon in Lowland Tropical Forests", the authors present data from a soil warming and drought experiment on low-fertility tropical rainforest soils. The data show that warming leads to a loss of older carbon, and more carbon over all, while drying decreased decomposition of recent inputs. These data are presented clearly and the interpretations are well-supported without being speculatively. Generally, this is an excellent manuscript that should require little editing before publication.

The main questions/issues/ideas that persist are

1) Why was the soil warming +4° C?

Author response:

Thank you for the overall positive comments. On this specific point, +4° C soil warming is consistent with climate projections for the increase in global surface and soil temperatures by the year 2100 reported by the IPCC RCP8.5 scenario (IPCC 2013, 2018) and the most recently named “SSP3-7.0” scenario (IPCC 2021). Tropical soils have been predicted to undergo up to 5° C warming (Soong et al). This level of warming (+4° C) is consistent with other soil warming studies that use the same approaches to achieve whole profile warming (Hicks Pries et al. 2017; Soong et al. 2021), and it is now the established benchmark warming treatment being deployed in a global network of deep soil warming experiments, in order to ensure comparable results across studies (Deep soil 2100); also see: <https://ucnrs.org/soil-warming-experiment-to-measure-future-carbon-emissions%EF%BF%BC/>.

This question made us realize that we had mistakenly identified the Hicks Pries et al., 2017 reference as 4.5°C when it should have been 4°C. This has been fixed in Line 234, so that it now reads “Other experiments that warmed the soil profile (by 4–4.5 °C)...”

Reviewer 2:

2) How were radiocarbon data interpreted? Was this simply by fitting data to estimate year of synthesis? Or by using a one-pool model? These provide slightly different answers, with the latter being more reliable. Please explain in the methods.

Author response:

The radiocarbon data were analyzed and interpreted as $\Delta^{14}\text{C}$ values as described in the Methods and reported in the Results. We provided approximate age differences for reference and ease of interpretation and specify in the text that they are approximate. The approximate age shift of 2–3 was determined by fitting data to estimate the average year of synthesis (fixation from the atmosphere) using the annual decline in atmospheric $\Delta^{14}\text{C}$ of 4.6‰ per year since 1995, as explained in the Supplementary Fig. 1 caption, which is referenced on Line 484. We have also clarified this in the methods of the main text (Lines 651-661).

We prefer this approach to using a one-pool model for several reasons. 1) These models rely on single year reported values for atmospheric ^{14}C . This is somewhat problematic for our sites because the reported value for 2019 is 0‰ (Hua et al. 2021), while we report an average value for $\Delta^{14}\text{C}$ $^{14}\text{CO}_2$ from air collected at our field sites of -4 ± 3 ‰. Because our differences in $\Delta^{14}\text{C}$ $^{14}\text{CO}_2$ are small, this discrepancy is problematic. We do not have local atmospheric values of $\Delta^{14}\text{C}$ $^{14}\text{CO}_2$ for the years prior to 2019 and the updated atmospheric calibration dataset is not yet published. 2) The one-pool model assumes steady state, and a two pool non-steady state simple model requires a timeseries of data, which we do not have. Considering these are climate manipulation experiments that are altering soil CO_2 flux and the sites near the Panama Canal are known to be depositional, it is reasonable to suspect that they are not at steady state. We expect to keep collecting data at these experiments and hope to be able to better model ages, turnover times, and transit times in the future. 3) We did run a one-pool steady state model for our dataset and assessed the

differences in ages determined this way to be 1-3 years, quite similar to our reported 2-3 years based on fitting to the year of fixation from the atmosphere. The difference of 1 year may be based on the Hua et al., 2021 calibration curve compared to the forecasted decline we used in our paper. The additional complexity of describing and defending the use of a single-pool steady-state model with a problematic calibration curve has led us to avoid this approach in favor of a much simpler and easier to understand calculation.

Reviewer 2:

3) Perhaps there could be a brief discussion of how the loss of old C and accumulation of younger plant-derived C shifts the overall SOM pool towards less stabilized C that is even more vulnerable to decomposition. This is mentioned in the abstract and conclusion but not in the main text.

Author response:

We thank the reviewer for identifying that this connection was unclear in the Discussion in the original manuscript. We have made additional changes to the Results and Discussion that we think make this clearer throughout. In addition, the original manuscript's Conclusion section is now the final paragraph of the Discussion (following Nature Communications formatting), thereby in the process following the reviewer's useful suggestion.

Reviewer 2:

Below are line-by-line comments, which are mostly minor.

L167-169: This sentence is a bit misleading. I believe the intention is "... the turnover of older decadal-aged carbon was less affected by drying than *was* the turnover of recent plant carbon inputs." The way it is written, it seems that the turnover of recent inputs more strongly affected the turnover of older C. Another approach could be "... the turnover of recent plant inputs was more strongly affected than older C to drying."

Author response:

Yes, we see that this was confusing. Thank you for pointing this out. We have revised this paragraph significantly to better convey our thoughts about mechanisms behind our findings and this sentence has been deleted. See Lines 341-354.

Reviewer 2:

L214: If the warming increases C loss (that is older) but drying reduces respiration while preserving younger C (through reduced utilization of inputs), then indeed the combined effects would probably be that C lost is older. This is addressed in the next paragraph, but my initial read of this sentence led me to question the result. Perhaps this can be slightly re-ordered to keep the reader from questioning the reasoning, possibly just by re-assessing the word "thus".

Yes, we see the reviewer's point. We have revised this section to more clearly develop this idea. See Lines 442-481.

Reviewer 2:

L220: This sentence was slightly confusing due to splitting the two effects. Consider adding “both”: “... warming and drying together may decrease both inputs of fresh carbon to soils as well as degradation rates...”

Author response:

Yes, good suggestion – but we have changed this sentence as well (see Lines 442-472).

Reviewer #3 (Remarks to the Author):

The authors demonstrated that soil carbon in tropical forest ecosystems is vulnerable to increased losses from heating and drying. The pathway to losses from heating centered on destabilization of decadal cycling soil C pools, while drying appeared to decrease stabilization of recent soil C inputs. The net impact of these two mechanisms indicate that the combination of heating and drying is likely to lead to a decrease in soil C sink strength for tropical forests. The novel approach taken in this study was to combine data from both heating and drying experiments in situ, relying on the power of radiocarbon measurements of respired CO₂. Using the bomb curve to inform the interpretation of the data points clearly to a decadal cycling pool as a prime contributor to soil C losses, while pairing the radiocarbon data with measurements of CO₂ production permits identification of mechanistic differences driving these responses under heating versus drying conditions. The authors effectively use the published literature along with their data to test mechanistic hypotheses, resulting in an important contribution to the field. Radiocarbon is a powerful tool, but interpretation of the data can be challenging. Putting the $\Delta^{14}\text{C}$ values observed in the context of the atmosphere for the year of sampling would be a useful point of reference. For example, some of the observations of total soil $\Delta^{14}\text{C}$ respired during the wet season at the SWELTR site are less than 0‰ in both the control and treatment sites, indicating that priming of centennially cycling soil C may be occurring during the wet season regardless of treatment. The inclusion of the atmospheric $\Delta^{14}\text{C}$ record for the bomb-C period in the supplemental information is helpful in this regard, but additionally mentioning the $\Delta^{14}\text{C}$ of the current atmosphere in the main text would be helpful. For example, in lines 102:105 the authors suggest that warming during the initial phase of the SWELTR experiment may have depleted more recent soil C inputs in the warmed plots but not the control plots. However, the $\Delta^{14}\text{C}$ measurements made during the dry season are closer to the atmosphere in the warmed plots than the control plots, suggesting potentially greater contributions of recently fixed soil C in the treated plots (although the differences are not statistically significant).

Author response:

Thank you – yes, ^{14}C is a powerful tool, but the interpretation of ^{14}C data can be challenging. In the revision, we incorporate the $\Delta^{14}\text{C}$ of air samples collected during our 2019 sampling

campaigns (4 +/- 3 permil) with the values shown as a reference line and described in the figure captions (Figures 2a and 3a). Regarding the trend toward ¹⁴C values closer to the atmosphere in the warming plots vs the control plots in the dry season: the p-value for the warming treatment effect in the dry season is 0.24, too high to consider marginally significant even if using an alpha level of 0.1. As such, we reiterate that there is no detected effect of warming on the ¹⁴C value of respired CO₂ in the dry season, when flux rates are relatively low.

Reviewer 3:

Inferring exact ages from radiocarbon observations alone requires a model, as soil is an open and heterogeneous system, i.e., continually receiving new C inputs while simultaneously producing CO₂ emissions (Trumbore, 2000, “Age of soil organic matter and soil respiration: radiocarbon constraints on belowground C dynamics”). Although a model is required to ascribe ages, observations of relatively enriched Δ¹⁴C values in soil C emissions from treated versus control plots is a clear indication that the primary source of emissions in the treated plots is C that was fixed after the bomb-C spike. The authors do acknowledge that soil C pools with a range of ages may be contributing to the observed fluxes, which is an important caveat for this analysis.

Author response:

Thank you, we provided approximate ages for reference and ease of interpretation and specify in the text that they are approximate. Our statistics and conclusions are based on the ¹⁴C isotopic values, not ages. We have more clearly described how we derived the ages in the Methods (Lines 651-661). We did apply a single pool steady state model to derive ages of respired C and the differences in these modeled ages did not differ widely from the ages based on time since fixation from the atmosphere. Since the modeled ages were derived using a model with a considerable amount of assumptions (steady state, a single pool homogenous with respect to likelihood of loss, a 1 year time lag for plant C inputs to enter the soil pool, and dependence on the published atmospheric ¹⁴C calibration curve which deviates from our measured values in 2019), we thought it better to use the simpler derivation in the manuscript, especially considering that the differences in ages are really provided for reference only.

Reviewer 3:

Estimating differences in soil C transit times in response to treatment would be one approach for obtaining a more quantitative estimate of the treatment effect on soil C cycling rates; this could be accomplished using a simple SoilR type model (Sierra et al., 2014, “Modeling radiocarbon dynamics in soils: SOILR version 1.1”; Stoner et al., 2021, “Soil organic matter turnover rates increase to match increased inputs in grazed grasslands”). Such a model could also be used as a conceptual tool to test the hypotheses advanced here, even if the data were not available for adequately constraining soil C pool sizes and turnover rates (Beem-Miller et al., 2021, “Impacts of drying and rewetting on the radiocarbon signature of respired CO₂ and implications for incubating archived soils”). For

example, it would be interesting to assess whether there is a difference in the effect on transit times for drying versus heating. Such a difference seems possible given that a) $\Delta^{14}\text{C}$ of respired CO_2 in the dry treatment plots of the PARCHED site was more enriched than that observed in the warmed plots of the SWELTR site, and b) the difference in treatment effects on CO_2 fluxes between the two experiments. Demonstrating differences in transit times are a clear way to estimate future soil C storage (cf. Sierra and Crow, 2022, “The climate benefit of sequestration in soils for warming mitigation”). However, the data are clear as is, and admittedly, such a modeling exercise would introduce new uncertainties, so may be outside the scope of the present work.

Author response:

Thank you for this suggestion. We spent some time considering how to implement this approach, especially considering that we do have additional ^{14}C data (see new Supplementary Figures 3 and 6) that can be used to initialize a model – at least for the P12 and SWELTR sites, which could be fit to the same model based on their similarities. We found that a relatively simple SoilR model (a three-pool parallel model) was able to produce ^{14}C of soil pools and laboratory incubations that matched observations of ^{14}C from soil pools and CO_2 from laboratory incubations well under “control” conditions. This model was able to approximate a drying treatment response similar to what we observed in the field by decreasing the k value of the particulate organic matter pool. However, we found it challenging to use the same model to capture a warming treatment response and found the best scenario was to decrease the k value of the mineral associated fraction while decreasing the proportion of inputs entering the particulate organic matter pool. When implemented together, these effects counteracted one another resulting in no change in the mean transit time of soil organic matter, which we found to be 8.5 years. This simple model did not fit measured stocks and fluxes well under control conditions, nor could we model ^{14}C values of CO_2 similar to our field data, which we worry might cause readers to doubt the fidelity of this modeling approach and detract from our field results. However, we thank the reviewer for this suggestion and will consider options for better capturing the impacts of warming and drying on soil C cycling with a model, possibly with a different implementation of SoilR or a different C model, in the future.

Reviewer 3:

Heterogeneity of soils is a major challenge for in situ warming experiments (and indeed, in soil organic matter science generally). The fact that the results in this study were robust in spite of this heterogeneity are promising. The authors point to the increase in heterogeneity observed during the wet season at SWELTR as potential evidence for priming of older soil organic matter decomposition alongside more recent soil C inputs, which is compelling when set alongside the references to existing literature on the importance of priming in nutrient limited systems such as these, and the observed increase in CO_2 fluxes. Vertical gradients in soil conditions are another key aspect of heterogeneity, and the authors posit that the decreasing impact of soil moisture with depth could be driving an increase in the relative contribution of respired CO_2 from deeper soil layers in the dry plots of the

PARCHED experiment. This seems plausible given the decrease in respiration and the increase in the apparent age of respired C.

Author response:

Thank you, we think so too!

Reviewer 3:

The authors pay special attention to seasonality, which is a major driver of soil CO₂ emissions in these ecosystems. Focusing on the time of the year with the greatest expected treatment effect clearly proved effective for testing the mechanistic hypotheses advanced in this study. However, it would have been helpful to put the data from this study in the context of the annual losses due to treatment observed for the sites, e.g., referencing the numbers provided by Nottingham et al., 2020 (“Soil carbon loss by experimental warming in a tropical forest”) for SWELTR, and given the decrease in respiration observed in response to the drying treatment, an annual estimate for PARCHED, e.g., Cusack et al., 2023, “Soil Respiration Responses to Throughfall Exclusion Are Decoupled From Changes in Soil Moisture for Four Tropical Forests, Suggesting Processes for Ecosystem Models”. This seems to be a missed opportunity for comparing the relative impacts of warming and drying (and the potential combined impact), which as the authors point out, will be critical for improving our understanding of the impacts of climate change on soil C cycling in tropical forest ecosystems.

Author response:

We can clearly place the time points used for this study as exemplifying the longer-term treatment effects by citing the papers reporting these data in the manuscript. We did miss the opportunity in the original manuscript draft to do so for PARCHED, where our data are consistent with the five-year treatment effect of suppressed soil CO₂ fluxes in drying plots versus controls during the dry and/or transitional seasons at these sites. We added this (Lines 315-318). We have also added references to other results from these experiments (mainly in response to another reviewer comment, see Lines 75, 192-222, 315-318, 338-380).

We think the reviewer’s suggestion to use annual fluxes to conceptually combine warming and drying effects might best be evaluated with a model that can simulate the combined impacts of warming and drying on soil carbon cycling. This is outside of the scope of the current manuscript, but it something we will consider for the future when longer post-treatment measurement periods and additional data better support this modeling activity.

Reviewer 3:

Additional notes:

Fig. S2: “dry” panel mislabeled as “NA” and panels are missing letters.

Author response:

This has been fixed.

Reviewer 3:

Line 174: But what is the mechanism for which the older C would remain accessible, while new C inputs are not? Increased contributions from deeper soil C, which are less affected by soil moisture, seems more plausible.

Author response:

Thanks for highlighting the need to add clarity on the potential mechanisms. We have addressed this in the revision in Lines 341-381. Briefly, we agree with the reviewer that differences in soil moisture constraints to respiration rates between surface and deep soils is a plausible mechanism to describe this result, as is described in lines. In addition, we further propose that moisture effects may also arise by moisture limitations to the initial breakdown and mobilization of fresh litter C (e.g., production and leaching of DOC from the litter layer down into the mineral soil or initial breakdown of subsurface root litter). This seems plausible considering buildup of litter and reduced litter decomposition rates with throughfall exclusion have been reported and water extractable C declined with throughfall exclusion at our sites. This might cause microbes to shift to locally accessible but less palatable C substrates. Without depth-resolved CO₂ production rates or depth-resolved ¹⁴CO₂ that demonstrate treatment effects, we do not have data to identify which is more likely – perhaps it is both. We hope that our continued work at these experiments will elucidate the answer! For the time being, we have discussed additional 14C data for our sites to expand on the potential mechanisms (See new Supplementary Figures 3 and 6 and Lines 218-232 and Lines 391-428). These data suggest, with some simplifying assumptions regarding the distribution of transit times in the different soil fractions and extrapolating across our study sites, that increased Δ¹⁴C values in surface soil CO₂ efflux with warming or drying could be explained by increased decomposition of surface occluded light fraction, surface dense fraction, subsurface light fraction, or deep roots.

References:

- Chacon, S. S., D. F. Cusack, A. Khurram, M. Bill, L. H. Dietterich, and N. J. Bouskill. 2023. 'Divergent responses of soil microorganisms to throughfall exclusion across tropical forest soils driven by soil fertility and climate history', *Soil Biology & Biochemistry*, 177: 35.
- Cusack, D. F., L. H. Dietterich, and B. N. Sulman. 2023. 'Soil Respiration Responses to Throughfall Exclusion Are Decoupled From Changes in Soil Moisture for Four Tropical Forests, Suggesting Processes for Ecosystem Models', *Global Biogeochemical Cycles*, 37.
- Dietterich, Lee H., Nicholas J. Bouskill, Makenna Brown, Biancolini Castro, Stephany S. Chacon, Lily Colburn, Amanda L. Cordeiro, Edwin H. García, Adonis Antonio Gordon, Eugenio Gordon, Alexandra Hedgpeth, Weronika Konwent, Gabriel Oppler, Jacqueline Reu, Carley Tsiamas, Eric Valdes, Anneke Zeko, and Daniela F. Cusack. 2022. 'Effects of experimental and seasonal drying on soil microbial biomass and nutrient cycling in four lowland tropical forests', *Biogeochemistry*, 161: 227-50.

- Hicks Pries, Caitlin E, Edward A. G Schuur, Susan M Natali, and K. Grace Crummer. 2016. 'Old soil carbon losses increase with ecosystem respiration in experimentally thawed tundra', *Nature Climate Change*, 6: 214-18.
- Hicks Pries, Caitlin E., C. Castanha, R. C. Porras, and M. S. Torn. 2017. 'The whole-soil carbon flux in response to warming', *Science*, 355: 1420-23.
- Hua, Quan, Jocelyn C. Turnbull, Guaciara M. Santos, Andrzej Z. Rakowski, Santiago Ancapichún, Ricardo De Pol-Holz, Samuel Hammer, Scott J. Lehman, Ingeborg Levin, John B. Miller, Jonathan G. Palmer, and Chris S. M. Turney. 2021. 'Atmospheric Radiocarbon for the Period 1950–2019', *Radiocarbon*: 1-23.
- IPCC. 2013. *Climate Change 2013: The Physical Science Basis. Contribution of Working Group I to the Fifth Assessment Report of the Intergovernmental Panel on Climate Change* (Cambridge University Press: Cambridge, United Kingdom New York, NY, USA).
- . 2018. "Global warming of 1.5°C. An IPCC Special Report on the impacts of global warming of 1.5°C above pre-industrial levels and related global greenhouse gas emission pathways,." In, edited by V. Masson-Delmotte, P. Zhai, H.-O. Pörtner, D. Roberts, J. Skea, P.R. Shukla, A. Pirani, W. Moufouma-Okia, C. Péan, R. Pidcock, S. Connors, J.B.R. Matthews, Y. Chen, X. Zhou, M.I. Gomis, E. Lonnoy, T. Maycock, M. Tignor, and T. Waterfield.
- . 2021. "Climate Change 2021: The Physical Science Basis. Contribution of Working Group I to the Sixth Assessment Report of the Intergovernmental Panel on Climate Change." In, edited by V. Masson-Delmotte, P. Zhai, A. Pirani, S.L. Connors, C. Péan, Berger S., N. Caud, Y. Chen, L. Goldfarb, M.I. Gomis, M. Huang, K. Leitzell, E. Lonnoy, J.B.R. Matthews, T.K. Maycock, T. Waterfield, O. Yelekçi, Yu R. and B. Zhou.
- Melillo, J. M., S. D. Frey, K. M. DeAngelis, W. J. Werner, M. J. Bernard, F. P. Bowles, G. Pold, M. A. Knorr, and A. S. Grandy. 2017. 'Long-term pattern and magnitude of soil carbon feedback to the climate system in a warming world', *Science*, 358: 101-05.
- Nottingham, Andrew T., Patrick Meir, Esther Velasquez, and Benjamin L. Turner. 2020. 'Soil carbon loss by experimental warming in a tropical forest', *Nature*, 584: 234-37.
- Nottingham, Andrew T., Jarrod J. Scott, Kristin Saltonstall, Kirk Broders, Maria Montero-Sanchez, Johann Püspök, Erland Bååth, and Patrick Meir. 2022. 'Microbial diversity declines in warmed tropical soil and respiration rise exceed predictions as communities adapt', *Nature Microbiology*, 7: 1650-60.
- Phillips, C. L., K. J. McFarlane, D. Risk, and A. R. Desai. 2013. 'Biological and physical influences on soil ¹⁴CO₂ seasonal dynamics in a temperate hardwood forest', *Biogeosciences*, 10: 7999-8012.
- Schuur, Edward A. G., Caitlin Hicks Pries, Marguerite Mauritz, Elaine Pegoraro, Heidi Rodenhizer, Craig See, and Chris Ebert. 2023. 'Ecosystem and soil respiration radiocarbon detects old carbon release as a fingerprint of warming and permafrost destabilization with climate change', *Philosophical Transactions of the Royal Society A: Mathematical, Physical and Engineering Sciences*, 381: 20220201.
- Soong, Jennifer L., Cristina Castanha, Caitlin E. Hicks Pries, Nicholas Ofiti, Rachel C. Porras, William J. Riley, Michael W. I. Schmidt, and Margaret S. Torn. 2021. 'Five years of whole-soil warming led to loss of subsoil carbon stocks and increased CO₂ efflux', *Science Advances*, 7: eabd1343.
- Wilson, R. M., A. M. Hoppole, M. M. Tfaily, S. D. Sebestyen, C. W. Schadt, L. Pfeifer-Meister, C. Medvedeff, K. J. McFarlane, J. E. Kostka, M. Kolton, R. K. Kolka, L. A. Kluber, J. K. Keller, T. P. Guilderson, N. A. Griffiths, J. P. Chanton, S. D. Bridgman, and P. J. Hanson. 2016. 'Stability of peatland carbon to rising temperatures', *Nature Communications*, 7: 13723.
- Wilson, Rachel M., Natalie A. Griffiths, Ate Visser, Karis J. McFarlane, Stephen D. Sebestyen, Keith C. Oleheiser, Samantha Bosman, Anya M. Hoppole, Malak M. Tfaily, Randall K.

Kolka, Paul J. Hanson, Joel E. Kostka, Scott D. Bridgman, Jason K. Keller, and Jeffrey P. Chanton. 2021. 'Radiocarbon Analyses Quantify Peat Carbon Losses With Increasing Temperature in a Whole Ecosystem Warming Experiment', *Journal of Geophysical Research: Biogeosciences*, 126: e2021JG006511.

REVIEWER COMMENTS

Reviewer #1 (Remarks to the Author):

I appreciate the authors' thorough revision. However, I remain unconvinced that a 2-3 year difference in carbon age can be considered significant enough to draw conclusions and implications from. While I understand that the authors refer to "older" carbon rather than "OLD" carbon, typical carbon ages span centuries and millennia even in tropical ecosystems. Hence, a difference of 2-3 years appears to have minimal significance. The authors argue that carbon cycles rapidly within weeks, but this does not necessarily demonstrate the significance of a 2-3 year difference unless it can be shown that a substantial amount of carbon falls within this age range. In reality, it is more likely that the majority of carbon is significantly older, spanning multiple decades or even hundreds of years. Therefore, the impact of such a small age difference seems minimal.

The approach used to estimate the radiocarbon age is problematic. "...we determined an approximate age shift with experimental warming and drying by fitting data to estimate the average year of synthesis (fixation from the atmosphere) using an annual decline in atmospheric $\Delta^{14}\text{C}$ of 4.6‰ calculated from 1995 to 2019". A low radiocarbon abundance could mean two things: It could mean the radiocarbon is from recent atmospheric radiocarbon or it could indicate that it is from pre-bomb period. Simply estimating the age using the atmospheric radiocarbon curve from 1995 to 2019 is flawed.

Lines 30-32: but drying did not increase soil CO₂ emissions. Please also change the sentence to EITHER accelerating xxx OR reducing xxx

Lines 198-199: isn't the increase in radiocarbon due to less contribution by young, fresh substrate?

Reviewer #2 (Remarks to the Author):

As per my initial review, I found this manuscript to be compelling and well-written. I was glad to see my comments and those of the other reviewers addressed in well-reasoned responses. The conclusions are well founded and clearly and narratively presented. Thus, I further my decision that this manuscript is in good shape for publication. I only have a few minor editorial notes.

I also appreciate your thorough exploration of methodology for interpreting ¹⁴C as per your response to my first review. The method chosen was well-justified, and will hopefully be adopted by others in this position.

Line by line comments:

L63: Move "carbon" after "aged" ?

L67-68: Should this be "the release of old C (depleted in...)"?

L298: This should probably be “decreased” instead of “decreasing”.

L301: There are a lot of commas in this sentence; perhaps it can be rewritten. Or, move the comma after “experiments” to after “results” on line 300.

Reviewer #3 (Remarks to the Author):

I appreciate the authors' thorough responses to the reviewers' comments, mine included. Specific noted improvements include:

- adding measurements of the $\Delta^{14}\text{C}$ of the atmosphere at the time of sampling helps to strengthen the mechanistic arguments made by the authors as well as improving the interpretability of Figs. 2a & 3a.
- additional explanation of the putative priming effect, e.g., through comparing the field respired $\Delta^{14}\text{C}$ to what was observed in laboratory incubations
- adding supplemental Figs. 3 & 6, which give depth-resolved $\Delta^{14}\text{C}$ values for heterotrophically respired CO_2 and bulk soils.
- enhanced interpretation of the observed variance in $\Delta^{14}\text{C}$ in reference to mechanistic hypotheses
- overall improvements in mechanistic explanations accompanied by additional supporting data for the presented hypotheses

Overall I am satisfied with the explanations and counterarguments from a scientific perspective, and therefore I think this is close to publication-ready. However, the manuscript could still benefit from minor grammatical changes to improve clarity. For example, the authors mix past and present tense in the results & discussion section, and there are several sentences that are overly convoluted or run-on. Additionally, some of the newly added text could be better integrated with the existing text, as it now reads with some redundancy. Together, these minor issues make it challenging to weigh the relative support of the many mechanistic hypotheses presented---at least I found myself getting lost at times, particularly with discussion of the effect of experimental drying. In sum, I think all of these issues could be easily remedied with another pass of minor editing, and I look forward to seeing this study published.

Final note: thanks for taking the time to address my question about SoilR modeling and transit times. Interesting findings!

Line-by-line notes:

L46: Suggest breaking this sentence into two.

Ln138-156: Mixed tense.

L199: Somewhat hard to follow this sentence. Maybe consider modifying the final clause with something like, "observations consistent with the putative change in substrate availability and use"?

L203: The cited reference argues that nutrient, rather than energetic, limitations are the primary constraint on decomposition in these soils. Does the enzymatic data support this?

L218: Maybe this new section should be a separate paragraph?

L221: consider, "when high decomposition rates *would be expected to* result in priming..."

L224-228: My reading of the relationship between bulk and respired $\delta^{14}\text{C}$ from SI Fig. 3 (nice data!) is that the majority of SOM in the upper mineral soil layers is cycling on decadal time scales. Critically, in the presence of fresh inputs (with $\Delta^{14}\text{C}$ near 0 per mil), this is the pool of SOM that dominates the CO_2 efflux, while in the absence of fresh inputs (lab incubations), the $\delta^{14}\text{C}$ of CO_2 efflux matches that of fresh plant inputs. Obviously this is your reading too, but it doesn't quite come across here. One suggestion is to separate the statements that (A) fine root biomass and root inputs are concentrated in the upper mineral soils, and (B) the majority of SOM in the upper mineral soils is cycling on decadal time scales.

L228: This sentence is a concise encapsulation of the findings, but reads as redundant here. Maybe move it up in the text?

L230: Define SOM

L239: delete "with"

L255: consider replacing "and" with "but"

L259-273: Consider splitting this into two sentences, e.g., "Soil CO_2 $\Delta^{14}\text{C}$ values decreased from the wet-to-dry season transition in May to the late wet season in November/December by 6 ± 3 ‰, when averaged across sites and treatment ($p < 0.01$). This may reflect the use of newer carbon (with $\Delta^{14}\text{C}$ closer to current atmospheric values) during the dry-to-wet season transition, possibly due to increased soil moisture enabling the decomposition of litter accumulated during the previous dry season."

L330: Not quite following here. Do you mean something like, "...drying decreases soil moisture, and has been shown to limit both microbial activity and the transport of [dissolved?] organic matter, as demonstrated following...?"

L334: Do you need "from the surface"? I find this sentence confusing as written.

L337: Consider replacing em dash with ", while"

L343: Consider "...soil moisture is higher, meaning that microbial activity is potentially less affected by changes in soil moisture, with the result that CO_2 produced at depth may comprise a larger component of total surface soil CO_2 efflux".

L335: New paragraph? Also, not clear which two cases you are talking about here.

L373: This mechanism (reduced fine root inputs) is introduced as a new mechanism, but seems to have already been discussed. Maybe this section could be rearranged to improve the flow?

L418: consider "...suggesting that this deep carbon pool is not contributing to our observation of increased $\Delta^{14}\text{C}$ of surface soil CO_2 efflux in the dry-to-wet season transition relative to the wet season, or with experimental drying relative to controls, at the SL site."

L425-426: consider "effect...attenuates"

L630: Not exactly true, e.g., same $\delta^{14}\text{C}$ values when bomb peak coming up as coming down.

Perhaps focus on the ability to distinguish $\delta^{14}\text{C}$ on annual to sub-annual resolution during the bomb C draw down period?

SI Fig. 1: Consider showing inset for viewers to better discern variation in measured data.

Point-by-point author responses are in italics.

REVIEWER COMMENTS

Reviewer #1 (Remarks to the Author):

I appreciate the authors' thorough revision. However, I remain unconvinced that a 2-3 year difference in carbon age can be considered significant enough to draw conclusions and implications from. While I understand that the authors refer to "older" carbon rather than "OLD" carbon, typical carbon ages span centuries and millennia even in tropical ecosystems. Hence, a difference of 2-3 years appears to have minimal significance. The authors argue that carbon cycles rapidly within weeks, but this does not necessarily demonstrate the significance of a 2-3 year difference unless it can be shown that a substantial amount of carbon falls within this age range. In reality, it is more likely that the majority of carbon is significantly older, spanning multiple decades or even hundreds of years. Therefore, the impact of such a small age difference seems minimal.

Author Response:

Thank you for recognizing our efforts in the revised manuscript. Our results reflect a shift in carbon substrates used for microbial respiration and released back to the atmosphere. A shift in C source with warming and drying is an important finding for tropical forests, which have the highest productivity and shortest turnover times for carbon of all terrestrial ecosystems. Tropical terrestrial systems have an average turnover time for carbon of 15 years from a global analysis (Carvalhais et al. 2014) and 6 to 18 years in a recent study in one tropical forest in the Amazon (Chanca et al. 2024). Our conclusion that these data indicate a shift on microbial substrate utilization is supported by related work in these experiments on microbial dynamics, soil biogeochemistry, and annual soil respiration fluxes (Cusack, Dietterich, and Sulman 2023; Chacon et al. 2023; Dietterich et al. 2022; Nottingham et al. 2022; Nottingham et al. 2020).

Others have reported similarly short soil carbon turnover and mean transit times for tropical forests. In a tropical forest in Puerto Rico with similar ^{14}C depth profiles to our sites, the mean transit time for 0-10 cm bulk soils was 27 years, increasing to 170 years (not thousands of years) at 60 cm depth (Mayer, McFarlane, and Silver 2024). More importantly, the mean pool age for light density fractions (10-25% of the total C pool depending on depth) were 1-4 years for the entire studied profile to 60 cm depth (Mayer, McFarlane, and Silver 2024), demonstrating the presence of an important and very rapidly cycling soil carbon pool at depth. A comparison of seven Neotropical lowland forests reported mean turnover times of less than 10 years for soils sampled to 10 cm depth (Posada and Schuur 2011). Turnover times of 10 years or less were reported soils for sampled to 22 cm depth from an Amazon forest in Brazil (Trumbore 1993). While turnover times of decades to millennia were reported for clay-associated soil C at depth, the

majority of soil C was found to have turnover times of less than a decade in Oxisols and Ultisols in Amazonian forests (Telles et al. 2003).

Our data show that we also observe similarly young soil carbon at our sites in Panama (Figures S3 and S6). Our density fractionations of soil from P12 and SL averaged 12% of total soil C in the 0-10 cm recovered in the free light fraction – at 25-50 cm depth, 7% of total soil C was recovered in the free light fraction. In preparing our initial response to Reviewer 3, who suggested we determine turnover times using the SoilR model (Sierra, Müller, and Trumbore 2014), we derived an estimate for the mean transit time of 8.5 years for soil C in the 0-10 cm depth increment for P12 and SWT. A similar model applied to SL yielded a mean transit time of 10.5 years for soil C in the 0-10 cm depth. Assumptions and parameterizations become more challenging to justify without a time series of data for subsoils.

Thus, we conclude that a shift in age of 2-3 years after only 1-3 years of experimental treatments is important considering how quickly soil cycles in tropical forest soils. To emphasize the rapid cycling of C in tropical systems, the second sentence of our Introduction now reads: “Tropical terrestrial ecosystems also have the shortest mean residence time for carbon on Earth, as short as 6–15 years (Carvalhais et al. 2014; Chanca et al. 2024), meaning that any change in carbon inputs or outputs could have large and relatively rapid consequences for tropical ecosystem carbon balance.” We have also added the following to the Discussion: “Others have reported soil carbon turnover times of less than 10 years for numerous tropical forests (Posada and Schuur 2011; Trumbore 1993). While turnover times of decades to millennia were reported for clay-associated soil C at depth, the majority of soil C was found to have turnover times of less than a decade in Oxisols and Ultisols in Amazonian forest (Telles et al. 2003). In forest in Puerto Rico, the mean carbon pool age for light density fractions (10–25% of the total C pool depending on depth) were 1–4 years to 60 cm depth (Mayer, McFarlane, and Silver 2024), demonstrating the presence of an important and very rapidly cycling soil carbon pool even at depth.” See Lines 36-38 and 327-333 on p16 of the revision pdf).

The approach used to estimate the radiocarbon age is problematic. “...we determined an approximate age shift with experimental warming and drying by fitting data to estimate the average year of synthesis (fixation from the atmosphere) using an annual decline in atmospheric $\Delta^{14}\text{C}$ of 4.6‰ calculated from 1995 to 2019”. A low radiocarbon abundance could mean two things: It could mean the radiocarbon is from recent atmospheric radiocarbon or it could indicate that it is from pre-bomb period. Simply estimating the age using the atmospheric radiocarbon curve from 1995 to 2019 is flawed.

All age estimates based on ^{14}C values face methodological assumptions and flaws. This is why we report and perform statistical analyses on the ^{14}C values and focus our discussion on changes in the ^{14}C values, rather than ages. As described in the paper, we provide the difference in ages for reference and ease of interpretation. We do not provide absolute ages in this manuscript because of the nuances behind the models used to derive ages

from ^{14}C data. We draw from our responses to Reviewer 2 and 3 from the initial review to further explain our reasoning for using this approach in deriving the difference in age with treatment below. We also provide the following caveat in the Methods (Line 454-456): “The $\Delta^{14}\text{C}$ of respired CO_2 can indicate an average age, or mean time elapsed since that carbon was fixed from the atmosphere, although it is important to recognize that this carbon is not homogenous in source or age.” And in the Discussion (Lines 320-323 on p16): “Importantly, our reported increases in ^{14}C with experimental warming and drying reflect changes in the average age of carbon being respired (the equivalent of 2–3 years in the mean age of respired carbon, see Supplementary Fig. 1) but do not provide insight into the distribution of carbon ages contributing to these averages.”

We agree it is possible that a decrease in ^{14}C values could mean that there is carbon from the pre-bomb period. We also hedge our description of the shift in ^{14}C observed seasonally in L217 for this reason. However, we observed increases (not decreases) in ^{14}C with treatment and our observations of higher ^{14}C in CO_2 produced during incubations and in shallow bulk soils and fractions further suggests that we are interpreting our results appropriately – that warming and drying treatments have shifted microbial C sources towards an increased proportion of C from slightly older (and higher ^{14}C abundance) soil carbon pools. The results of our turnover and transit time modeling also support our interpretation. In addition to the 3-pool model applied with SoilR, which confirms the shorter turnover time is appropriate, we fit our respiration ^{14}C data to a one-pool model as an alternative approach to estimating the age of respired C following Vaughn and Torn, 2019. As the reviewer may be aware, the one-pool model provides two solutions for ^{14}C values above current atmosphere – one corresponding to the increase in the atmospheric ^{14}C - CO_2 curve and one corresponding to the decreasing side of the curve. The one-pool model yielded an average age across all sites, treatments, and seasons of 5 years for the shorter turnover time and 243 years for the longer turnover time solution, which seems highly unlikely considering these are tropical soils in highly productive seasonally moist forests, the ^{14}C of CO_2 was higher in laboratory incubations than in the field, and the average ^{14}C values of the surface soils and fractions (and several subsurface bulk soil pools and fractions) are also higher than those of soil CO_2 efflux *in situ*.

Finally, we prefer our approach to using a one-pool model for estimating the age of respired C for several reasons. 1) Turnover time models rely on single year reported values for atmospheric ^{14}C . This is somewhat problematic for our sites because the reported value for 2019 is 0‰ (Hua et al. 2022), while we report an average value for $\Delta^{14}\text{C}$ of CO_2 from air collected at our field sites of -4 ± 3 ‰. Because our differences in $\Delta^{14}\text{C}$ of CO_2 are small, this discrepancy is problematic. We do not have local atmospheric values of $\Delta^{14}\text{C}$ of CO_2 for the years prior to 2019 and the updated atmospheric calibration dataset is not yet published. 2) The one-pool model assumes steady state, and a two pool non-steady state simple model requires a timeseries of data, which we do not have. Considering these are climate manipulation experiments that are altering soil CO_2 flux and the sites near the Panama Canal are known to be depositional, it is reasonable to suspect that they are not at steady state. We expect to keep collecting data at these experiments and hope to be able

to better model ages, turnover times, and transit times in the future. 3) We did run a one-pool steady state model for our dataset and assessed the differences in ages determined this way to be 1-3 years, quite similar to our reported 2-3 years based on fitting to the year of fixation from the atmosphere. The difference of 1 year may be based on the Hua et al., 2021 calibration curve compared to the forecasted decline we used in our paper, which better matches our observed value for ^{14}C of CO_2 in local air. The additional complexity of describing and defending the use of a single-pool steady-state model with a problematic calibration curve has led us to avoid this approach in favor of a much simpler and easier to understand calculation, especially considering the resulting difference in age is the same using both approaches. We provided approximate ages for reference and ease of interpretation and specify in the text that they are approximate. Our statistics and conclusions are based on the ^{14}C isotopic values, not ages.

Lines 30-32: but drying did not increase soil CO_2 emissions. Please also change the sentence to EITHER accelerating xxx OR reducing xxx

Author Response:

We see your point and have changed “and” to “or”. We have also changed “both” to “each in Line 5 to increase clarity that treatments were not applied together. Other minor edits to the abstract also make it more clear that warming increased CO_2 emissions, but drying decreased CO_2 emissions. See Lines 25-30 of the Abstract.

Lines 198-199: isn't the increase in radiocarbon due to less contribution by young, fresh substrate?

Author Response:

Yes, we suspect this is the mechanism behind this increase in ^{14}C . This sentence has been revised as follows: “We found that experimental drying led to an increase in the mean $\Delta^{14}\text{C}$ of respired CO_2 by $8 \pm 3 \text{‰}$ averaged across sites and sampling periods ($p = 0.03$, Fig. 3a), consistent with a putative shift in microbial substrate use towards older, decadal-aged soil carbon.” See Lines 205-207 on p10. Additional discussion of this mechanism follows two paragraphs below.

Reviewer #2 (Remarks to the Author):

As per my initial review, I found this manuscript to be compelling and well-written. I was glad to see my comments and those of the other reviewers addressed in well-reasoned responses. The conclusions are well founded and clearly and narratively presented. Thus, I further my decision that this manuscript is in good shape for publication. I only have a few minor editorial notes.

I also appreciate your thorough exploration of methodology for interpreting $\delta^{14}\text{C}$ as per your response to my first review. The method chosen was well-justified, and will hopefully be adopted by others in this position.

Author Response:

Thank you for your comments. We are pleased that you are satisfied with our review. We have made the requested edits as described below.

Line by line comments:

L63: Move “carbon” after “aged’ ”? *Done.*

L67-68: Should this be “the release of old C (depleted in...”? *Yes, we have added “carbon” here. Thank you.*

L298: This should probably be “decreased” instead of “decreasing”. *Yes, we have changed “decreasing” to “decreased”. Thank you.*

L301: There are a lot of commas in this sentence; perhaps it can be rewritten. Or, move the comma after “experiments” to after “results” on line 300.

Author Response:

Yes, we have edited this sentence and agree it is easier to read as follows: “Thus, while further full factorial experiments are needed to elucidate the combined effects of warming and drying on soil carbon and its release as soil CO_2 efflux, our results indicate that warming and drying together will increase losses of older and previously stable soil carbon.” See Lines 307-310 on p16.

Reviewer #3 (Remarks to the Author):

I appreciate the authors' thorough responses to the reviewers' comments, mine included. Specific noted improvements include:

- adding measurements of the $\delta^{14}\text{C}$ of the atmosphere at the time of sampling helps to strengthen the mechanistic arguments made by the authors as well as improving the interpretability of Figs. 2a & 3a.
- additional explanation of the putative priming effect, e.g., through comparing the field respired $\delta^{14}\text{C}$ to what was observed in laboratory incubations
- adding supplemental Figs. 3 & 6, which give depth-resolved $\delta^{14}\text{C}$ values for heterotrophically respired CO_2 and bulk soils.
- enhanced interpretation of the observed variance in $\delta^{14}\text{C}$ in reference to mechanistic hypotheses

- overall improvements in mechanistic explanations accompanied by additional supporting data for the presented hypotheses

Author Response:

Thank you for the positive feedback. We agree that these changes have improved the manuscript.

Overall I am satisfied with the explanations and counterarguments from a scientific perspective, and therefore I think this is close to publication-ready. However, the manuscript could still benefit from minor grammatical changes to improve clarity. For example, the authors mix past and present tense in the results & discussion section, and there are several sentences that are overly convoluted or run-on. Additionally, some of the newly added text could be better integrated with the existing text, as it now reads with some redundancy. Together, these minor issues make it challenging to weigh the relative support of the many mechanistic hypotheses presented---at least I found myself getting lost at times, particularly with discussion of the effect of experimental drying. In sum, I think all of these issues could be easily remedied with another pass of minor editing, and I look forward to seeing this study published.

Author Response:

We appreciate this feedback. We found it somewhat challenging to incorporate the new data into the manuscript and follow your suggestions to improve flow and clarity with minor edits throughout the paper. We have made several other minor edits for improved clarity throughout the manuscript. We have edited tenses to more consistently present our results in the past tense throughout the Results and Discussion. We have reorganized the section discussing the effects of experimental drying. We agree that the flow of this section was difficult to follow and have rearranged it so that now it begins with a possible reduction of new plant C into soils (both litter and roots). Then the alternative mechanism, that CO₂ production moved down the soil profile is presented. We also moved the references to the soil and soil fraction 14C data earlier in this section. See lines 247-285 on pages 13-14.

Final note: thanks for taking the time to address my question about SoilR modeling and transit times. Interesting findings!

Author Response:

Thank you!

Line-by-line notes:

L46: Suggest breaking this sentence into two. *Done.*

Ln138-156: Mixed tense. *This has been fixed.*

L199: Somewhat hard to follow this sentence. Maybe consider modifying the final clause

with something like, "observations consistent with the putative change in substrate availability and use"?

Author Response: This has been changed to: "We found that experimental drying led to an increase in the mean $\Delta^{14}\text{C}$ of respired CO_2 by 8 ± 3 ‰ averaged across sites and sampling periods ($p = 0.03$, Fig. 3a), consistent with a putative change in substrate use and availability."

L203: The cited reference argues that nutrient, rather than energetic, limitations are the primary constraint on decomposition in these soils. Does the enzymatic data support this?

Author response: Thank you for the question. Dietterich et al., 2022 reported several enzyme activities for the four PARCHED sites and found higher enzymatic activities and lower forest floor biomass at the high fertility site (P13) compared to the low fertility sites (P12, SL, and Gigante). They also found no consistent seasonal effects on enzyme activities. So yes, these data support the idea that nutrient limitations are the primary constraint on decomposition in these soils. However, microbial biomass was lower in the late dry than wet season at SL and in the early wet than late wet season at P12 (Dietterich et al. 2022). As we only worked in low fertility sites and focus on seasonal effects in this sentence, it does not seem necessary to include the enzyme activities here in the manuscript. This paper is also referenced for the seasonal patterns in microbial biomass later in Line 298.

L218: Maybe this new section should be a separate paragraph? Yes, thank you. We have made it a separate paragraph.

L221: consider, "when high decomposition rates *would be expected to* result in priming..." Good suggestion, we've made this change.

L224-228: My reading of the relationship between bulk and respired ^{14}C from SI Fig. 3 (nice data!) is that the majority of SOM in the upper mineral soil layers is cycling on decadal time scales. Critically, in the presence of fresh inputs (with $\Delta^{14}\text{C}$ near 0 per mil), this is the pool of SOM that dominates the CO_2 efflux, while in the absence of fresh inputs (lab incubations), the ^{14}C of CO_2 efflux matches that of fresh plant inputs. Obviously this is your reading too, but it doesn't quite come across here. One suggestion is to separate the statements that (A) fine root biomass and root inputs are concentrated in the upper mineral soils, and (B) the majority of SOM in the upper mineral soils is cycling on decadal time scales.

Author Response:

Yes, this is our interpretation of these data. We thought it simpler to focus on the high C inputs from plants and the presence of more ^{14}C in these layers (which if decomposed could increase the ^{14}C values of the CO_2 efflux). We appreciate the suggestion to take this further and state that the majority of C in these shallow layers is cycling quickly. We have

split this sentence into shorter statements to make these points more clearly and have made additional changes to this section in response to other Reviewer suggestions.

L228: This sentence is a concise encapsulation of the findings, but reads as redundant here. Maybe move it up in the text?

Author Response:

We have moved this up to in the latest revision, to what is now the preceding paragraph. We have also edited the following sentence to improve clarity.

L230: Define SOM *Done*, but at first use (Line 155) with “soil organic matter” replaced with SOM elsewhere in the manuscript.

L239: delete "with" *Done*.

L255: consider replacing "and" with "but" *Done*.

L259-273: Consider splitting this into two sentences, e.g., "Soil CO₂ Δ 14C values decreased from the wet-to-dry season transition in May to the late wet season in November/December by 6 ± 3 ‰, when averaged across sites and treatment ($p < 0.01$). This may reflect the use of newer carbon (with Δ 14C closer to current atmospheric values) during the dry-to-wet season transition, possibly due to increased soil moisture enabling the decomposition of litter accumulated during the previous dry season." *Done*.

L330: Not quite following here. Do you mean something like, "...drying decreases soil moisture, and has been shown to limit both microbial activity and the transport of [dissolved?] organic matter, as demonstrated following..."?

Author response:

Not quite, it has been challenging to directly link decreases in soil moisture to decreases in soil respiration in throughfall exclusion experiments – likely because of the ways soil moisture is most often measured (limited number of depths or large depth increments, not enough sensors to detect small but ecologically significant changes, etc.). We have revised section based on other suggestions and this sentence has been modified to more clearly discuss our findings.

L334: Do you need "from the surface"? I find this sentence confusing as written.

Author Response:

We agree it is unnecessary and have removed “from the surface”.

L337: Consider replacing em dash with ", while" *Done*.

L343: Consider "...soil moisture is higher, meaning that microbial activity is potentially less

affected by changes in soil moisture, with the result that CO₂ produced at depth may comprise a larger component of total surface soil CO₂ efflux".

Author Response:

We have modified this sentence as suggested and further edited it for consistency in tense. See Lines 272-274.

L335: New paragraph? Also, not clear which two cases you are talking about here.

Author Response:

Yes, we see why this was unclear. This section has been revised to more clearly convey our interpretation of our data and the potential underlying mechanisms. We have split this paragraph so that the seasonal effects on CO₂ flux rates is discussed in a separate section summarizing the trends across all three sites.

L373: This mechanism (reduced fine root inputs) is introduced as a new mechanism, but seems to have already been discussed. Maybe this section could be rearranged to improve the flow?

Author Response:

Thank you for the suggestion. We agree that the flow of this section was difficult to follow and have rearranged it so that now it begins with a possible reduction of new plant C into soils (both litter and roots). Then the alternative mechanism, that CO₂ production moved down the soil profile is presented. See Lines 270-285 on p14.

L418: consider "...suggesting that this deep carbon pool is not contributing to our observation of increased $\Delta^{14}\text{C}$ of surface soil CO₂ efflux in the dry-to-wet season transition relative to the wet season, or with experimental drying relative to controls, at the SL site." *Done.*

L425-426: consider "effect...attenuates" *Done.*

L630: Not exactly true, e.g., same ¹⁴C values when bomb peak coming up as coming down. Perhaps focus on the ability to distinguish ¹⁴C on annual to sub-annual resolution during the bomb C draw down period?

Author Response:

We have specified this in the Radiocarbon Interpretation section. See Line 450.

SI Fig. 1: Consider showing inset for viewers to better discern variation in measured data. *Done.*

- Carvalhais, Nuno, Matthias Forkel, Myroslava Khomik, Jessica Bellarby, Martin Jung, Mirco Migliavacca, Mingquan Mu, Sassan Saatchi, Maurizio Santoro, Martin Thurner, Ulrich Weber, Bernhard Ahrens, Christian Beer, Alessandro Cescatti, James T. Randerson, and Markus Reichstein. 2014. 'Global covariation of carbon turnover times with climate in terrestrial ecosystems', *Nature*, 514: 213-17.
- Chacon, S. S., D. F. Cusack, A. Khurram, M. Bill, L. H. Dietterich, and N. J. Bouskill. 2023. 'Divergent responses of soil microorganisms to throughfall exclusion across tropical forest soils driven by soil fertility and climate history', *Soil Biology & Biochemistry*, 177: 35.
- Chanca, I., I. Levin, S. Trumbore, K. Macario, J. Lavric, C. A. Quesada, A. Carioca de Araújo, C. Quaresma Dias Júnior, H. van Asperen, S. Hammer, and C. Sierra. 2024. 'How long does carbon stay in a near-pristine central Amazon forest? An empirical estimate with radiocarbon', *EGUsphere*, 2024: 1-23.
- Cusack, D. F., L. H. Dietterich, and B. N. Sulman. 2023. 'Soil Respiration Responses to Throughfall Exclusion Are Decoupled From Changes in Soil Moisture for Four Tropical Forests, Suggesting Processes for Ecosystem Models', *Global Biogeochemical Cycles*, 37.
- Dietterich, Lee H., Nicholas J. Bouskill, Makenna Brown, Biancolini Castro, Stephany S. Chacon, Lily Colburn, Amanda L. Cordeiro, Edwin H. García, Adonis Antonio Gordon, Eugenio Gordon, Alexandra Hedgpeth, Weronika Konwent, Gabriel Oppler, Jacqueline Reu, Carley Tsiames, Eric Valdes, Anneke Zeko, and Daniela F. Cusack. 2022. 'Effects of experimental and seasonal drying on soil microbial biomass and nutrient cycling in four lowland tropical forests', *Biogeochemistry*, 161: 227-50.
- Hua, Q., J. C. Turnbull, G. M. Santos, A. Z. Rakowski, S. Ancapichún, R. De Pol-Holz, S. Hammer, S. J. Lehman, I. Levin, J. B. Miller, J. G. Palmer, and C. S. M. Turney. 2022. 'Atmospheric Radiocarbon for the Period 1950-2019', *Radiocarbon*, 64: 723-45.
- Mayer, A. C., K. J. McFarlane, and W. L. Silver. 2024. 'The effect of repeated hurricanes on the age of organic carbon in humid tropical forest soil', *Glob Chang Biol*, 30: e17265.
- Nottingham, Andrew T., Patrick Meir, Esther Velasquez, and Benjamin L. Turner. 2020. 'Soil carbon loss by experimental warming in a tropical forest', *Nature*, 584: 234-37.
- Nottingham, Andrew T., Jarrod J. Scott, Kristin Saltonstall, Kirk Broders, Maria Montero-Sanchez, Johann Püspök, Erland Bååth, and Patrick Meir. 2022. 'Microbial diversity declines in warmed tropical soil and respiration rise exceed predictions as communities adapt', *Nature Microbiology*, 7: 1650-60.
- Posada, Juan, and Edward Schuur. 2011. 'Relationships among precipitation regime, nutrient availability, and carbon turnover in tropical rain forests', *Oecologia*, 165: 783-95.
- Sierra, C. A., M. Müller, and S. E. Trumbore. 2014. 'Modeling radiocarbon dynamics in soils: SoilR version 1.1', *Geosci. Model Dev.*, 7: 1919-31.
- Telles, Everaldo de Carvalho Conceição, Plínio Barbosa de Camargo, Luiz A. Martinelli, Susan E. Trumbore, Enir Salazar da Costa, Joaquim Santos, Niro Higuchi, and Raimundo Cosme Oliveira, Jr. 2003. 'Influence of soil texture on carbon dynamics and storage potential in tropical forest soils of Amazonia', *Global Biogeochem. Cycles*, 17: 1040.

Trumbore, S.E. 1993. 'Comparison of carbon dynamics in tropical and temperate soils using radiocarbon measurements', *Global Biogeochemical Cycles*, 7: 275-90.

REVIEWERS' COMMENTS

Reviewer #1 (Remarks to the Author):

Thank you to the authors for the detailed explanation and clarification. I'm satisfied with the revisions. Congratulations on such a great piece of work.